# PROVABLE IN-CONTEXT LEARNING OF NONLINEAR REGRESSION WITH TRANSFORMERS

## ABSTRACT

The transformer architecture has revolutionized machine learning by processing input sequences into outputs. A defining feature is in-context learning (ICL)—the ability to perform unseen tasks from prompts without updating model parameters. Early theoretical work focused on linear tasks, and recent studies have begun exploring nonlinear functions. Yet a rigorous analysis of the training dynamics—how transformers learn such complex tasks—remains elusive. This paper presents the first formal analysis of ICL training dynamics for a broad class of nonlinear regression functions. We analyze the stage-wise dynamics of attention during training: attention scores between a query token and its target features rise rapidly at first, then gradually converge to one, while attention to irrelevant features decays more slowly and can oscillate. Our analysis explicitly characterizes how general non-degenerate $L$-Lipschitz task functions shape attention weights, identifying the Lipschitz constant $L$ as the key factor governing the convergence dynamics. Leveraging these insights, for two distinct regimes depending on whether $L$ is below or above a threshold, we derive different time bounds to guarantee near-zero prediction error. Despite convergence time depending on the task, we prove query tokens ultimately focus on highly relevant prompt tokens, demonstrating transformers' robust ICL capability.

## 1 INTRODUCTION

The transformer architecture (Vaswani et al., 2017) has driven transformative advances across a wide spectrum of machine learning domains, including computer vision (Bi et al., 2021; Han et al., 2022; Goldblum et al., 2024), natural language processing (Kalyan et al., 2021; Tunstall et al., 2022), and speech processing (Mehrish et al., 2023; Latif et al., 2023). A salient feature of transformers is their ability to perform new tasks without updating parameters, simply by conditioning on a few input-output examples—known as prompts. This capability, referred to as *in-context learning (ICL)*, enables models to generalize to unseen tasks purely through inference (Brown et al., 2020).

ICL has attracted growing interest, with numerous empirical studies examining when transformers succeed or fail at in-context generalization (Xie et al., 2021; Garg et al., 2022; Von Oswald et al., 2023; Wu et al., 2023; Li et al., 2024b; Agarwal et al., 2024; Park et al., 2025). Notably, Garg et al. (2022) provided preliminary theoretical evidence that transformers trained on specific function classes (e.g., linear) can accurately infer a query function value from prompts containing variable–function pairs, highlighting transformers' surprising ability to "learn" within their forward pass and mimic classical function approximation.

Building on this foundation, subsequent works have provided *theoretical* understandings of ICL by characterizing the training dynamics of single-layer attention transformers (Mahankali et al., 2023; Zhang et al., 2024; Huang et al., 2024; Collins et al., 2024; Yang et al., 2024). For instance, Huang et al. (2024) consider softmax attention to analyze how attention weights evolve during training linear regression problems. More recent studies have theoretically shown that transformers can learn specific nonlinear function classes in-context, such as binary classification, low-degree polynomial regression, and Gaussian single-index models (Li et al., 2024a; Yang et al., 2024; Oko et al., 2024; Sun et al., 2025). However, these studies do not provide a full theoretical picture of how the step-by-step learning process is governed by the task itself. To date, a formal characterization of the pre-training dynamics for general nonlinear ICL has been a key open problem.

In this work, we take a step toward understanding the *learning dynamics* for ICL on a broad class of nonlinear regression functions. We address two fundamental questions: (1) Which geometric properties of the target function govern the convergence behavior of transformer-based ICL? and (2) Despite nonlinearity and generality, how can a transformer learn in context to achieve low prediction error? We answer both by analyzing transformer training under gradient descent. Our main contributions are summarized below.

- *Broad Class of Nonlinear Functions and Flexible Feature Sets:* Our analysis generalizes previous studies in two ways. (i) Unlike prior theoretical works that focus on linear mappings (Zhang et al., 2024; Huang et al., 2024), binary classification (Li et al., 2024a), or low-degree polynomials (Sun et al., 2025), we characterize learning dynamics for a much broader family of non-degenerate $L$-Lipschitz task functions without assuming low complexity. This class is satisfied by almost any nontrivial $L$-Lipschitz function (e.g., scaled cosine, piecewise-polynomial, and small neural-network functions) and excludes only degenerate cases. (ii) Our results also hold for general feature embeddings without the restrictive orthonormality assumptions in prior work (Huang et al., 2024; Li et al., 2024a; Chen et al., 2024; Nichani et al., 2024).

- *Phase Transition of Training Dynamics between Flat and Sharp Curvature Regimes:* We discover a phase transition in *training dynamics* governed by the Lipschitz constant $L$. When $L$ is below a threshold of order $\Theta\left(\frac{1}{\Delta\delta}\right)$, the **flat curvature regime** yields smaller gradients and permits larger step sizes to converge. When $L$ exceeds the threshold, the **sharp-curvature regime** produces larger gradients requiring smaller steps. The two regimes exhibit distinct convergence behaviors: although the flat regime may converge faster at high accuracy, sufficiently large $L$ enhances feature separability, enabling accelerated training in the sharp regime.

- *Convergence Guarantee for ICL with Nonlinear Regression Functions:* We provide formal convergence guarantees for a one-layer softmax attention model learning nonlinear regression functions. We prove that gradient descent achieves near-zero training loss in polynomial time across both flat and sharp $L$-regimes. We also characterize the two-phase training dynamics: an early phase in which attention scores between query tokens and target features rise rapidly, and a later phase in which these scores converge to one while attention to irrelevant features decays more slowly with oscillations. At convergence, query tokens consistently attend to highly relevant prompt tokens, demonstrating the ICL capability of transformers.

- *Novel Analysis Techniques:* We develop new proof tools that explicitly connect the curvature of nonlinear task functions to the evolution of attention weights. In particular, we first decompose the prediction loss to explicitly relate it to attention weights and cross-feature gaps under nonlinear functions. We further show that the flat and sharp curvature regimes of the parameter $L$ lead to distinct gradient magnitudes, which in turn drive different convergence rates and shape the overall training dynamics. The impact of function curvature on the magnitude and stability of attention updates is, to our knowledge, not addressed in previous ICL literature (Huang et al., 2024; Oko et al., 2024; Cheng et al., 2024).

## 2 RELATED WORKS

**In-context learning.** In-context learning (ICL) has emerged as a fundamental capability of transformer models, enabling dynamic task adaptation without parameter updates. The research community has approached ICL from several distinct yet complementary perspectives.

The Bayesian inference perspective, pioneered by Xie et al. (2021) and further developed by Zhang et al. (2023); Wang et al. (2023); Falck et al. (2024), establishes a theoretical framework linking prompting strategies to probabilistic reasoning. This line of work interprets ICL as a form of implicit Bayesian model averaging, where transformers effectively perform approximated inference conditioned on the provided context. Another line of ICL work focuses on Markov chains to study the behavior of induction heads for transformers, which are designed to copy or compare tokens that follow previous occurrences (Nichani et al., 2024; Bietti et al., 2023; Edelman et al., 2024; Rajaraman et al., 2024). The central focus is to understand how transformers recover latent sequence structures (Nichani et al., 2024) and transition rules (Ren & Liu, 2024; Li et al., 2023; Makkuva et al., 2024), using token-level recurrence and dynamics.

In contrast, the function learning perspective initiated by Garg et al. (2022) demonstrated transformers' remarkable ability to learn and interpolate simple function classes (particularly linear models) directly from context examples. This work sparked significant interest in understanding the mechanistic underpinnings of ICL, leading to important discoveries by Von Oswald et al. (2023) and Dai et al. (2022), who revealed deep connections between attention mechanisms and gradient-based optimization dynamics. Recent advances have substantially expanded our understanding of ICL mechanisms: Akyürek et al. (2023) provided a rigorous analysis of linear regression tasks, showing that trained transformers can implement both ridge regression gradient descent and exact least-squares solutions. Bai et al. (2023) established comprehensive theoretical results encompassing expressive power, prediction capabilities, and sample complexity, while proposing general mechanisms for algorithmic selection. Cheng et al. (2024) interprets transformers as implementing functional gradient descent for nonlinear regression, but their analysis primarily focuses on the representational capacity and functional viewpoint of the learned predictor, without analyzing the ICL pre-training dynamics.

**Theoretical analysis of ICL learning dynamics.** Recent theoretical work has made significant progress in understanding the pre-training dynamics of ICL in transformers, though important limitations remain (Huang et al., 2024; Li et al., 2024a; Collins et al., 2024; Yang et al., 2024; Sun et al., 2025; Lu et al., 2024; Edelman et al., 2024; Lin & Lee, 2024; Jeon et al., 2024; Park et al., 2025). The foundational work by Huang et al. (2024) established the first rigorous analysis of training dynamics for softmax attention in ICL settings, focusing on a single-head attention layer learning linear regression tasks. Their key theoretical result demonstrates that prompt tokens with features identical to the query token develop dominating attention weights during training. However, this analysis relies critically on strong assumptions about pairwise orthonormality of feature vectors and a normalized function scale, limiting its applicability to more general settings. Our work goes beyond this setting by showing that, for general nonlinear targets, the Lipschitz constant of the underlying function governs both the gradient evolution and the resulting convergence regimes. In this sense, our analysis includes the linear case as a special instance while providing a more fine-grained understanding of how function curvature influences attention-based learning.

Subsequent work extended these results to classification tasks (Li et al., 2024a), dual-head settings (Lin & Lee, 2024), and structured or hierarchical functions (Yang et al., 2024; Sun et al., 2025). Some studies explored the implicit bias of gradient descent and attention-based generalization (Collins et al., 2024; Lu et al., 2024), while others examined information-theoretic and dynamical aspects of ICL (Edelman et al., 2024; Jeon et al., 2024). Despite these advances, most analyses still rely on restrictive assumptions, such as orthonormality features, fixed positional encoding, or carefully structured input distributions, limiting their ability to explain ICL under general tasks and practical learning conditions.

**Notations.** In this paper, for a vector $v$, we let $\|v\|_2$ denote its $\ell$-2 norm. For some positive constant $C_1$ and $C_2$, we define $x = \Omega(y)$ if $x > C_2|y|$, $x = \Theta(y)$ if $C_1|y| < x < C_2|y|$, and $x = \mathcal{O}(y)$ if $x < c_1|y|$. We also denote by $x = o(y)$ if $x/y \to 0$. We use $\mathrm{poly}(C)$ to denote large constant degree polynomials of $C$. For a matrix $A$, we use $A_i$ to denote the $i$-th column of $A$, and $A_{i:j}$ to represent the collection of columns from the $i$-th to the $j$-th column (inclusive).

## 3 SYSTEM MODEL

In this section, we formulate our system model, including the problem setup for in-context learning, the transformer architecture, and the associated training process.

### 3.1 IN-CONTEXT LEARNING PROBLEM SETUP

We consider a standard in-context learning (ICL) framework commonly used in prior studies (Garg et al., 2022; Huang et al., 2024; Yang et al., 2024). The objective is to train a transformer model that can perform ICL over a designated class of functions $\mathcal{F}$, where each function $f \in \mathcal{F}$ corresponds to one task context. Here, we focus on *nonlinear* function classes further elaborated below.

Given each task (i.e., given one function $f$ randomly sampled from $\mathcal{F}$), a prompt with a sequence of $N$ input-response pairs $(x_i, y_i)$ as well as a query input $x_{\text{query}}$ are sampled, where $x_i \in \mathcal{X} \subseteq \mathbb{R}^d$, and $y_i = f(x_i)$. Let the input matrix $X = (x_1 \quad x_2 \quad \cdots \quad x_N) \in \mathbb{R}^{d \times N}$ and the response vector $\mathbf{y} = (y_1 \quad y_2 \quad \cdots \quad y_N) \in \mathbb{R}^{1 \times N}$. We adopt the following standard prompt embedding (Garg et al.,

2022; Huang et al., 2024; Yang et al., 2024):

$$P = \begin{pmatrix} x_1 & x_2 & \cdots & x_N & x_{\text{query}} \\ y_1 & y_2 & \cdots & y_N & 0 \end{pmatrix} = \begin{pmatrix} X & x_{\text{query}} \\ \mathbf{y} & 0 \end{pmatrix} \in \mathbb{R}^{(d+1)\times(N+1)}. \tag{1}$$

**Non-degenerate $L$-Lipschitz Regression Functions.** In this work, we focus on regression tasks, where each task is associated with a regression function drawn from the following set

$$\mathcal{F} = \left\{ f : \begin{array}{l} |f(x) - f(x')| \leq L\|x - x'\|, \quad \forall \|x - x'\| = \Theta(\delta_0), \\ \forall v_k \in \mathbb{V}, \exists v_{k'} \in \mathbb{V}, \ k' \neq k, \ \text{such that } |f(v_k) - f(v_{k'})| = \Theta(L) \cdot \|v_k - v_{k'}\| \end{array} \right\}, \tag{2}$$

where $L > 0$ and $\delta_0 = \mathcal{O}(1)$. In particular, the functions in the class are said to satisfy **non-degenerate $L$-Lipschitz** condition, which imposes two natural requirements on the function class. First, the standard global $L$-Lipschitz condition ensures that the function $f$ does not change too rapidly, which is common in the literature and includes a wide class of linear and nonlinear functions. Second, the separation requirement guarantees that for any feature $v_k$ in the set $\mathbb{V}$, there exists at least one other feature $v_{k'}$ such that the function difference between them achieves the order of variation defined by the Lipschitz constant. This ensures sufficient distinguishability of features by the function class for guaranteed learnability. This mild separation assumption is satisfied by almost any nontrivial $L$-Lipschitz function (e.g., scaled cosine, piecewise-polynomial, and small neural-network functions) and excludes only degenerate cases. Together, these two conditions ensure that the function class is sufficiently rich for ICL while avoiding unlearnable or trivial scenarios.

For each prompt $P$, the task-specific function $f(x)$ is independently drawn based on a task distribution $\mathcal{D}_f$, as long as $f(x)$ satisfies the property for the same $L$ and $\delta_0$ in Eq. (2).

**Feature Embeddings.** Let $\mathbb{V} := \{v_k \in \mathbb{R}^d | k = 1, \cdots, K\}$ be the feature embeddings of tokens. For any $k \neq k'$, we assume a separation of $\|v_k - v_{k'}\| = \Theta(\Delta)$, where $\Delta = \Theta(1)$. Each data sample $x$ is modeled as a noisy perturbation of one of the vectors in $\mathbb{V}$. This assumption lets us control the separation $\Delta$ precisely, simplifying the analysis while retaining the essential geometry of the problem. Such a condition can be satisfied by various feature learning techniques to avoid feature collapse, e.g., disentangled representation learning (Wang et al., 2022; 2024; Higgins et al., 2018). We note that such a condition substantially generalizes the orthonormality assumption taken by the previous study (Huang et al., 2024; Li et al., 2024a; Chen et al., 2024; Nichani et al., 2024). The prompt is sampled as follows. For a randomly chosen $v_k$, we assume $x$ satisfies $\|x - v_k\| = O(\epsilon_x)$ with probability $p_k$, where $\epsilon_x = o(1)$ and $p_k = \Theta(\frac{1}{K})$. For analytical simplicity, we assume $x = v_k$ whenever this proximity condition holds. In our experiments in Section 6, we further verify that our training dynamic analysis remains valid when tokens are drawn from general continuous distributions.

## 3.2 One-Layer Transformer

In this work, we adopt a one-layer transformer model for solving the ICL problem, which is commonly used in the existing theoretical ICL literature (e.g., Huang et al. (2024); Li et al. (2024a); Yang et al. (2024); Sun et al. (2025)). A self-attention transformer with width $d_e$ consists of a key matrix $W^K \in \mathbb{R}^{d_e \times d_e}$, a query matrix $W^Q \in \mathbb{R}^{d_e \times d_e}$, and a value matrix $W^V \in \mathbb{R}^{d_e \times d_e}$. For a given prompt $P$ of length $N$ in Eq. (1), the self-attention layer outputs:

$$F(P; W^K, W^Q, W^V) = W^V P \times \text{softmax}\left((W^K P)^\top W^Q P\right). \tag{3}$$

where the $\text{softmax}(\cdot)$ function is applied column-wisely, i.e., for a vector input $z$, the $i$-th entry of $\text{softmax}(z)$ is given by $\text{softmax}(z_i) = \frac{\exp(z_i)}{\sum_j \exp(z_j)}$.

We further take the following re-parameterization, commonly adopted by the recent theoretical studies of transformers (e.g. Zhang et al. (2024); Huang et al. (2024); Yang et al. (2024); Sun et al. (2025)), which combines the query and key matrices into a single matrix $W^{KQ} \in \mathbb{R}^{(d+1)\times(d+1)}$, and further specify the weight matrices as follows:

$$W^V = \begin{pmatrix} 0_{d\times d} & 0_d \\ 0_d^\top & 1 \end{pmatrix}, \quad W^{KQ} = \begin{pmatrix} Q & 0_d \\ 0_d^\top & 0 \end{pmatrix},$$

where $Q \in \mathbb{R}^{d\times d}$ is the trainable weight matrix. These simplifications, while not capturing the full complexity of deep, multi-head models, are standard in the theoretical literature and serve two crucial purposes. First, they allow for a tractable analysis that isolates the core dynamics of the softmax

attention mechanism, which is central to ICL. Second, this setup follows a line of foundational work that has successfully used similar models to derive key insights into ICL for linear regression. By building on this framework, we can directly investigate the impact of nonlinearity. We acknowledge that extending our analysis to multi-layer and multi-head settings is an important avenue for future work. Such structured matrices separate out the impact of inputs and responses and have been justified to achieve the global or nearly global optimum for both linear and softmax attention models in Zhang et al. (2024) and Huang et al. (2024). Then the self-attention mapping becomes

$$F(P;Q) = \mathbf{y} \cdot \text{softmax}\big(X^\top Q \bar{X}\big), \tag{4}$$

where we further let $\bar{X} = (x_1 \quad x_2 \quad \cdots \quad x_N \quad x_{\text{query}}) \in \mathbb{R}^{d \times N+1}$ and $\mathbf{y} = (y_1 \quad y_2 \quad \cdots \quad y_N) \in \mathbb{R}^{d \times N}$. The prediction $\hat{y}_{\text{query}}$ corresponding to $x_{\text{query}}$ is given by the last entry of $F(P;Q)_{N+1}$, i.e., $\hat{y}_{\text{query}} = F(P;Q)_{N+1}$. To train the attention model on the ICL problem introduced in Section 3.1, we minimize the following squared loss between the predicted and true responses:

$$\mathcal{L}(P;Q) = \tfrac{1}{2}\mathbb{E}\Big[\big(F(P;Q)_{N+1} - f(x_{\text{query}})\big)^2\Big], \tag{5}$$

where the expectation is taken over the randomly sampled prompt $\{x_i\}_{i=1}^N \cup \{x_{\text{query}}\}$ and randomly sampled function $f \in \mathcal{F}$ that determines the corresponding ground-truth responses.

We optimize this loss via gradient descent (GD). Let $\text{vec}(Q)$ denote the vector that stacks all entries of $Q$. At $t = 0$, we initialize $\text{vec}(Q)^{(0)}$ as the zero matrix $0_{d^2}$. The parameter is updated as follows:

$$\text{vec}(Q)^{(t+1)} = \text{vec}(Q)^{(t)} - \eta \nabla_{\text{vec}(Q)} \mathcal{L}\big(P; \text{vec}(Q)^{(t)}\big). \tag{6}$$

where $\eta > 0$ is the learning rate. Note that we require $\eta$ to be smaller than a universal constant (e.g., $\eta < 1$) to ensure stability of the update and to preserve the convergence behavior analyzed in Section 5. Based on this model setup and training procedure, we proceed to present our main theoretical results concerning ICL under nonlinear regression tasks.

## 4 MAIN RESULTS OF ICL CONVERGENCE

Our analysis proceeds by decomposing the loss function into interpretable quantities that directly reflect cluster separation and the Lipschitz constant $L$ in Eq. (2). Interestingly, we observe that the *flat* and *sharp* regimes of $L$ give rise to distinct convergence dynamics, with a threshold transition separating the two regimes based on the order of magnitude of $L$. In the flat regime (small $L$), the gradient on $Q$ matrices remains small, leading to slow but steady concentration of attention weights. In the sharp regime (large $L$), we show a rapid growth phase where the query–key inner products amplify differences between clusters before settling into a slow fine-tuning phase. For the two regimes, we provide explicit convergence-time bounds and characterize the phase transition. Compared with prior analyses restricted to linear tasks or orthogonal features, our framework extends to general nonlinear Lipschitz tasks and explains qualitatively different dynamics observed in practice.

Recall from Section 3.1 that each token $x_i$ corresponds to a noisy version of a feature $v_k \in \mathbb{V}$ with probability $p_k = \Theta\left(\frac{1}{K}\right)$ for any $k \in [K]$. Let $P_{1:N}$ denote the collection of input tokens in $P$, i.e., $\{x_i\}_{i=1}^N$, and denote $\mathcal{V}_k \subset [N]$ as the index set for input tokens, such that $x_i = v_k$ for $i \in \mathcal{V}_k$. We define the following concentration set of token sequences where each feature appears with approximately the expected frequency:

$$\mathcal{E}^* := \Big\{P_{1:N} : |\mathcal{V}_k| \in \big[(p_k - \delta)N, (p_k + \delta)N\big] \text{for } k \in [K]\Big\}, \tag{7}$$

where $\delta \geq \sqrt{\frac{20K}{N}}$. Then, for any $0 < \epsilon < 1$, suppose $N \geq \Theta(K^3)$ and $K \geq \Theta(\frac{1}{\epsilon})$. For any $t \in [T]$, we have the concentration probability satisfies $\mathbb{P}(P_{1:N} \in \mathcal{E}^*) \geq 1 - 3\exp\left(-\frac{\delta^2 N}{25}\right)$. This implies that, with high probability, each feature class $k \in [K]$ is approximately equally represented in the prompt $P$, ensuring a balanced token distribution. Such balance is crucial for the convergence of ICL, as it allows the attention to learn effectively from all feature types without introducing bias. While our analysis is expressed in terms of the population risk in Eq. (5), the concentration event in Eq. (7) ensures that the empirical prompt distribution closely matches the population distribution when $N$ is sufficiently large. Under this event, the curvature-driven attention dynamics characterized later in our theory remain accurate up to standard $\mathcal{O}(1/\sqrt{N})$ fluctuations. Thus, the population-level analysis offers a faithful description of the finite-sample training behavior for prompts of moderate size.

Given balanced feature inputs, we now quantify how much the query token $x_{\text{query}}$ attends to specific input tokens or feature classes. We define the attention score for a query token to attend to the $i$-th token in the prompt as $\text{attn}_i^{(t)} := \text{softmax}(x_i^\top Q^{(t)} x_{\text{query}})$ and the total attention paid to all tokens with feature $v_k$ as $\text{Attn}_k^{(t)} := \sum_{i:x_i=v_k} \text{attn}_i^{(t)}$. With this notation, the transformer's output at $t$ is

$$\hat{y}_{\text{query}} = \sum_{i \in [N]} \text{attn}_i^{(t)} y_i = \sum_{k \in [K]} \text{Attn}_k^{(t)} f(v_k). \tag{8}$$

Building on the expression in Eq. (8), we characterize how the attention scores influence the prediction loss defined in Eq. (5) in the following lemma.

**Lemma 1.** *Given constants $L, \Delta > 0$, the prediction loss in Eq. (5) can be expressed as:*

$$\mathcal{L}(P; Q) = \frac{1}{2} \sum_{k=1}^{K} \mathbb{E}\left[ \mathbb{1}\{x_{\text{query}} = v_k\} \left(1 - \text{Attn}_k^{(t)}\right)^2 \cdot \mathcal{O}(L^2 \Delta^2) \right], \; \forall t \in [T], \tag{9}$$

*where $\mathbb{1}\{x_{\text{query}} = v_k\} = 1$ is the indicator function that equals 1 if $x_{\text{query}} = v_k$ and 0 otherwise.*

The proof of Lemma 1 is given in Appendix C. This expression reveals that loss $\mathcal{L}(P; Q)$ depends on the Lipschitz constant $L$, the feature gap $\Delta$, and the attention score $\text{Attn}_k^{(t)}$ associated with the true feature of the query token. The dependence on $L$, which captures the function's curvature, distinguishes our analysis from prior theoretical work on ICL (e.g., Huang et al. (2024); Oko et al. (2024); Sun et al. (2025)). For a given feature gap $\Delta$, different function Lipschitz constants $L$ can lead to distinct convergence behaviors of ICL. In the following theorem, we first provide the $\epsilon^2$-convergence of $\mathcal{L}(P; Q)$ in the flat $L$-regime, where $L$ is below a certain threshold.

**Theorem 1** (Flat $L$-regime). *Suppose the function class in Eq. (2) satisfies $L \le \Theta\left(\frac{1}{\Delta \delta}\right)$. Then, for any $0 < \epsilon < 1$ and under $N \ge \Theta(K^3)$ and $K \ge \Theta(\frac{1}{\epsilon})$, with at most $T_f^* = \Theta(\frac{K \log(K \epsilon^{-1})}{\eta \delta^2 L^2 \Delta^2})$ iterations, we have $\mathcal{L}(P; Q) \le \mathcal{O}(\epsilon^2)$.*

The proof of Theorem 1 is given in Appendix E.3. Theorem 1 indicates that $T_f^*$ decreases (i.e., the convergence is faster) as either the Lipschitz constant $L$ or the feature gap $\Delta$ increases. Intuitively, a larger function Lipschitz constant $L$ implies that the outputs $y_i$ and $y_{i'}$ corresponding to different features in the prompt $P$ become more distinguishable, which facilitates faster in-context learning. Similarly, a larger feature gap $\Delta$ improves the separability among features, enabling the query token to more accurately attend to the relevant prompt tokens.

However, when $L$ exceeds a certain threshold, the caused sharp curvature induces large gradients on the attention weights. As a result, a smaller stepsize is required to stabilize convergence, leading to a convergence rate that differs from that in the flat regime. We next establish the $\epsilon^2$-convergence of $\mathcal{L}(P; Q)$ in the sharp $L$ regime, where $L$ is above the threshold.

**Theorem 2** (Sharp $L$-regime). *Suppose the function class in Eq. (2) satisfies $L = \Omega(\frac{1}{\Delta \delta})$. Then for any $0 < \epsilon < 1$, under $N = \Omega(K^3)$ and $K \ge \Theta(\frac{1}{\epsilon})$, with at most $T_s^* = \Theta(\frac{K \log(K \epsilon^{-1} L \Delta)}{\eta \epsilon \delta^2 L^2 \Delta^2})$ iterations, we have $\mathcal{L}(P; Q) = \mathcal{O}(\epsilon^2)$.*

The proof of Theorem 2 is given in Appendix F.2. Theorem 1 and Theorem 2 together reveal an interesting **phase transition phenomenon**: the convergence dynamics are governed by how the function Lipschitz constant $L$ compares to a threshold of order $\Theta\left(\frac{1}{\Delta \delta}\right)$. In the **flat curvature regime**, where $L$ is below this threshold, convergence allows larger step sizes due to smaller gradients, resulting in a convergence rate of $\tilde{\Theta}\left(\frac{K}{\eta \delta^2 L^2 \Delta^2}\right)$. In contrast, in the **sharp curvature regime**, where $L$ is above the threshold, the large $L$ incurs large gradients which thus require smaller step sizes to stabilize convergence, yielding a convergence rate of $\tilde{\Theta}\left(\frac{K}{\eta \epsilon \delta^2 L^2 \Delta^2}\right)$. Comparing the two convergence upper bounds, neither $T_f^*$ nor $T_s^*$ always dominates. The sharp regime benefits from a larger $L$, giving a smaller denominator and faster convergence, but its bound also contains an extra $\frac{1}{\epsilon}$ factor that can dominate when high accuracy is required. Therefore, depending on the relative scales of $L, \epsilon$, and $\Delta$, either regime may achieve the smaller convergence upper bound.

In cases where the query is not a small perturbation of any feature vector, the model must represent the query using a combination of multiple feature clusters rather than relying on a single dominant one. This setting is more challenging because the nonlinear function values evolve over training, causing the optimal attention pattern to shift across clusters. Nevertheless, the curvature-dependent

gradient behavior established in our analysis continues to govern the convergence rate, even though the precise attention trajectory becomes harder to characterize.

Based on the convergence results above, we now characterize the behavior of the attention score $\text{Attn}_k^{(t)}$ at convergence to explain why the ICL output corresponds to an accurate prediction.

**Proposition 1.** *After the prediction loss converges to $\mathcal{L}(P; Q) = \mathcal{O}(\epsilon^2)$ for any $0 < \epsilon < 1$, if the query token satisfies $x_{\text{query}} = v_k$, then the attention score associated with feature $v_k$ satisfies $1 - \text{Attn}_k^{(t)} = \mathcal{O}(\epsilon)$.*

The proof of Proposition 1 is given in Appendix G. This result follows directly from the loss expression in Eq. (9), where $\mathcal{L}(P; Q) = \Theta((1 - \text{Attn}_k^{(t)})^2)$ when $x_{\text{query}} = v_k$. Intuitively, as $\text{Attn}_k^{(t)}$ approaches 1, the attention matrix effectively focuses predominantly on the tokens that share the same feature $v_k$. As a result, the predicted output $\hat{y}_{\text{query}}$, given by Eq. (8), closely approximates the true value $f(v_k)$, leading to high-accuracy predictions.

## 5 ANALYSIS OF CONVERGENCE DYNAMICS

As established in Theorem 1 and Theorem 2, different regimes of the Lipschitz constant $L$ lead to distinct convergence behaviors in ICL. In this section, we analyze the training dynamics under both regimes, highlighting how the Lipschitz constant $L$ influences the convergence rate in each case.

### 5.1 GRADIENTS OF ATTENTION WEIGHTS

Based on the prediction output $\hat{y}_{\text{query}}$ in Eq. (5), the attention scores $\text{attn}_i$ play a critical role in determining the final prediction. To precisely characterize these attention scores for any $i \in [N]$, it is sufficient to characterize the training dynamics of the attention weights $q_{k,k'}^{(t)} := v_{k'}^\top Q^{(t)} v_k$ for $k, k' \in [K]$, which are initialized as $q_{k,k'}^{(0)} = 0$ for any $k, k' \in [K]$. To simplify notations, we denote the $q_{k,k}^{(t)}$ as $q_k^{(t)}$ for $k' = k$. According to the definition of the attention score $\text{attn}_i^{(t)}$ in Eq. (8), when $x_{\text{query}}^{(t)} = v_k$, the quantity $q_k^{(t)}$ measures how strongly the query token attends to the target feature $v_k$, while $q_{k,k'}^{(t)}$ reflects the attention given to a different feature $v_{k'}$ with $k' \neq k$. To achieve the desired attention behavior, effective training should increase $q_k^{(t)}$ while suppressing $q_{k,k'}^{(t)}$.

The convergence behavior of the transformer depends on the dynamics of $q_k^{(t)}$ and $q_{k,k'}^{(t)}$. We therefore proceed to analyze how these quantities evolve during training. To this end, we define the gradient updates for $q_k^{(t)}$ and $q_{k,k'}^{(t)}$ as $g_k^{(t)}$ and $g_{k,k'}^{(t)}$, respectively. Under gradient descent with learning rate $\eta$, the update rules are given by:

$$q_k^{(t+1)} := q_k^{(t)} + \eta g_k^{(t)}, \qquad q_{k,k'}^{(t+1)} := q_{k,k'}^{(t)} + \eta g_{k,k'}^{(t)}.$$

We now present the following lemma, which provides the exact expressions for the gradient terms $g_k^{(t)}$ and $g_{k,k'}^{(t)}$ for a function class $\mathcal{F}$ with Lipschitz constant $L$.

**Lemma 2.** *For any $t \in [T]$, suppose $x_{\text{query}} = v_k$. Then for any $k, k' \in [K]$ with $k' \neq k$, we obtain*

$$g_k^{(t)} = \mathbb{E}\left[\mathbb{1}\{x_{\text{query}} = v_k\}\text{Attn}_k^{(t)}\left(1 - \text{Attn}_k^{(t)}\right)^2 \cdot \Theta(L^2\Delta^2)\right], \tag{10}$$

$$|g_{k,k'}^{(t)}| = \mathbb{E}\left[\mathbb{1}\{x_{\text{query}}^{(t)} = v_k\}\text{Attn}_{k'}^{(t)} \cdot (1 - \text{Attn}_k^{(t)}) \cdot (1 - \text{Attn}_{k'}^{(t)}) \cdot \Theta(L^2\Delta^2)\right]. \tag{11}$$

The proof of Lemma 2 is given in Appendix D. From Eq. (10), we observe that $g_k^{(t)}$ is always non-negative, implying that the update $q_k^{(t)}$ increases over time. This growth continues until the attention score $\text{Attn}_k^{(t)}$ approaches its convergence state near 1. However, $g_{k,k'}^{(t)}$ in Eq. (11) is not necessarily positive and also depends on $\text{Attn}_{k'}^{(t)}$ associated with feature vector $v_{k'}$. However, as $\text{Attn}_k^{(t)}$ approaches 1, the residual term in Eq. (11) diminishes, and $g_{k,k'}^{(t)}$ also converges toward zero, facilitating the overall convergence of the system.

As also shown in Lemma 2, both gradients scale with the Lipschitz constant $L$ and the feature gap $\Delta$, illustrating their influence on the training dynamics. In the following subsections, we analyze how different regimes of the function Lipschitz constant $L$ (with respect to the threshold determined by $\Delta$) affect the evolution of $q_k^{(t)}$ and $q_{k,k'}^{(t)}$ through the gradients $g_k^{(t)}$ and $g_{k,k'}^{(t)}$, thereby offering deeper insight into the convergence results established in Theorem 1 and Theorem 2.

## 5.2 CONVERGENCE DYNAMICS UNDER FLAT $L$-REGIME

For ease of exposition, we consider the case where $x_{\text{query}} = v_k$ in the following. Under the initialization of $q_k^{(0)} = q_{k,k'}^{(0)} = 0$, and by the definition of the attention score, we have $\text{attn}_n^{(0)} = \frac{1}{N}$ for all $n \in [N]$, meaning that the transformer initially attends equally to all input tokens when computing the prediction for $x_{\text{query}}$. We then leverage the task distribution in Eq. (2) and the gradient expressions in Lemma 2 to analyze the learning dynamics of $q_k^{(t)}$ and $q_{k,k'}^{(t)}$.

In the initial phase of training, the prediction $\hat{y}_{\text{query}}$ is far from the ground truth $f(v_k)$ due to the zero initialization of bilinear weights. According to Eq. (10), this results in a large positive gradient $g_k^{(t)}$, leading to a rapid increase in $q_k^{(t)}$. In contrast, the gradient $g_{k,k'}^{(t)}$ may fluctuate in sign depending on the alignment of $f(v_k)$ and $f(v_{k'})$ at each step, causing $q_{k,k'}^{(t)}$ to oscillate but decrease much more slowly. We formally characterize this phase below.

**Proposition 2** (Phase I: Fast growth of $q_k^{(t)}$). *For any $t \in \{1, \cdots, T_f^1\}$ and $k \in [K]$, where $T_f^1 = \Theta(\frac{K \log(K)}{\eta L^2 \Delta^2})$, the attention weight $q_k^{(t)}$ increases at a rate of $\Theta(\frac{\eta L^2 \Delta^2}{K})$. Meanwhile, $q_{k,k'}^{(t)}$ oscillates at a slower rate of $\mathcal{O}(\frac{\eta L^2 \Delta^2}{K^2})$ and exhibits an overall decreasing trend. By the end of Phase I (i.e., $t = T_f^1 + 1$), we have $\text{Attn}_k^{(T_f^1+1)} = \Omega(\frac{1}{1+\delta})$.*

The proof of Proposition 2 is given in Appendix E.1. According to Proposition 2, Phase I ends once $q_k^{(t)}$ becomes sufficiently large to reduce the prediction gap between the ICL output and the ground truth function value. After this point, both $g_k^{(t)}$ in Eq. (10) and $|g_{k,k'}^{(t)}|$ in Eq. (11) decrease to smaller orders. During this phase, both $g_k^{(t)}$ and $|g_{k,k'}^{(t)}|$ increase with $L$, as a larger function Lipschitz constant induces greater residual values in Eq. (10) and Eq. (11). Likewise, a larger feature gap $\Delta$ amplifies the value difference between $f(v_k)$ and $f(v_{k'})$ (by Eq. (2)), which in turn accelerates attention learning in Phase I. As a result, the duration $T_f^1$ decreases with both $L$ and $\Delta$.

However, at $t = T_f^1$, the prediction loss in Eq. (5) may still remain non-negligible. As a result, the attention score $\text{Attn}_k^{(t)}$ requires a period of steady improvement after $t = T_f^1 + 1$. We now formalize this behavior in the second training phase in the following proposition.

**Proposition 3** (Phase II: Steady growth of $q_k^{(t)}$ under flat $L$-regime). *For any $t \in \{T_f^1 + 1, \cdots, T_f^*\}$ and $0 < \epsilon < 1$, where $T_f^* = \Theta(\frac{K \log(K\epsilon^{-1})}{\eta \delta^2 L^2 \Delta^2})$, for any $k \in [K]$, $q_k^{(t)}$ continues to grow at a steady rate of $\Theta\left(\frac{\eta \delta^2 L^2 \Delta^2}{K}\right)$. Meanwhile, $q_{k,k'}^{(t)}$ oscillates at a slower rate of $\mathcal{O}(\frac{\eta \delta^2 L^2 \Delta^2}{K^2})$ and exhibits an overall decreasing trend. At $t = T_f^* + 1$, if $L$ satisfies $L \leq \Theta(\frac{1}{\Delta \delta})$ in Eq. (2), we have $\text{Attn}_k^{(T_f^*+1)} = \Omega(\frac{1}{1+\epsilon\delta})$.*

The proof of Proposition 3 is given in Appendix E.2. According to Proposition 3, if the Lipschitz constant is sufficiently small such that $L \leq \Theta(\frac{1}{\Delta \delta})$, then by the end of Phase II, the attention score satisfies $1 - \text{Attn}_k^{(T_f^*+1)} = \mathcal{O}\left(\frac{\epsilon\delta}{1+\epsilon\delta}\right) = \mathcal{O}(\epsilon)$, indicating that the transformer has converged.

## 5.3 PHASE TRANSITION UNDER SHARP $L$-REGIME

In the sharp curvature regime where $L = \Omega(\frac{1}{\Delta \delta})$, the update $q_k^{(t)}$ increases at a rate of $\Theta(\frac{\eta}{K})$ for all $t \leq T_f^*$. Let $T_s^1$ denote the duration of Phase I in this regime. Since this early-stage growth (Phase I) mirrors the dynamics under the flat $L$-regime before $T_f^*$, we have $T_s^1 = T_f^* = \Theta(\frac{K \log(K\epsilon^{-1})}{\eta \delta^2 L^2 \Delta^2})$.

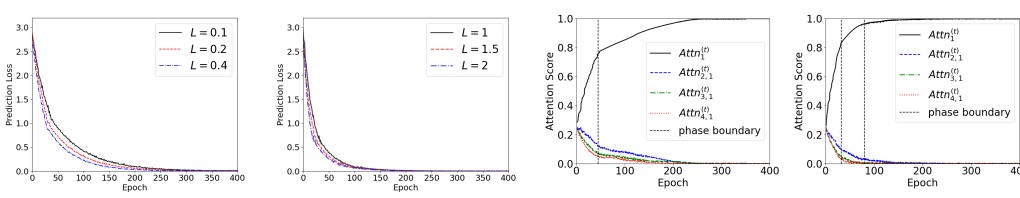

(a) Prediction loss for flat $L$-set. (b) Prediction loss for sharp $L$-set. (c) Attention scores for flat $L = 0.1$. (d) Attention scores for sharp $L = 1$.

Figure 1: Training dynamics of prediction losses (top) and attention scores (bottom) for two sets of $L$: flat ($\{0.1, 0.2, 0.4\}$) and sharp ($\{1.0, 1.5, 2.0\}$).

However, this initial Phase is insufficient for achieving convergence due to the residual error term proportional to $L \cdot \Delta$ in Eq. (9). Consequently, training transitions into a second phase (Phase III), during which both gradient terms $g_k^{(t)}$ and $g_{k,k'}^{(t)}$ become small. This leads to a slower growth rate of $q_k^{(t)}$ compared to the earlier phase. We characterize this slower training phase as follows.

**Proposition 4** (Phase III: Slow growth of $q_k^{(t)}$ under sharp $L$-regime). *If $L = \Omega(\frac{1}{\Delta\delta})$, then for any $t \in \{T_s^1 + 1, \cdots, T_s^*\}$ and $0 < \epsilon < 1$, where $T_s^* = \Theta(\frac{K \log(KL\Delta\epsilon^{-1})}{\eta\epsilon\delta^2 L^2\Delta^2})$, for any $k \in [K]$, $q_k^{(t)}$ increases at a rate of $\Theta(\frac{\eta\delta^2 L^2\Delta^2\epsilon}{K})$. Meanwhile, $q_{k,k'}^{(t)}$ fluctuates at a slower rate of $\mathcal{O}(\frac{\eta\delta^2 L^2\Delta^2\epsilon}{K^2})$. By the end of Phase II (i.e. $t = T_s^* + 1$), we have $\text{Attn}_k^{(T_s^* + 1)} = \Omega(\frac{1}{1+\epsilon\delta})$.*

The proof of Proposition 4 is given in Appendix F.1. To ensure convergence of the prediction loss, the update step size of $q_k^{(t)}$ decreases to an order dependent on $\epsilon$ in Phase II. This is because, under the sharp $L$-regime where $\delta^2 L^2\Delta^2 = \Omega(1)$, the gradients become large, as established in Lemma 2. This phase transition further supports the convergence guarantee in Theorem 2, demonstrating that even under sharp curvature, the attention mechanism gradually concentrates on the correct feature vector, ultimately enabling accurate prediction.

## 6 EXPERIMENTS VERIFICATION

We adopt the data and task distributions from Section 3.1. Each data point is sampled from a fixed feature set $v_k \in \mathbb{R}^d$, $k = 1, \ldots, K$, with each feature $v_k$ chosen uniformly at random, i.e., $p_k = 1/K$. Each task involves learning a cosine function of the form $f(x) = \frac{L}{c} \cdot \cos(c \cdot x)$, where $c > 0$ is a random constant and $L$ is the Lipschitz constant, satisfying Eq. (2). Each prompt consists of $N$ randomly sampled inputs $\{x_i\}_{i=1}^N$ and corresponding outputs $\{y_i\}_{i=1}^N = \{f(x_i)\}_{i=1}^N$, along with a query token $x_{\text{query}}$. We set the parameters as follows: $d = 15$, $K = 4$, $N = 100$, $c = 0.5$, and $\Delta = 3$. We generate $M = 300$ prompts and train the model for $T = 400$ epochs. Appendix B presents additional experiments, including attention map dynamics, robustness check with non-uniform feature frequencies, polynomial-function tasks, and deeper transformers.

We analyze a simplified transformer model comprising a single block with one-head self-attention and a feedforward network, incorporating layer normalization and ReLU activation, followed by a linear output layer. Our analysis focuses on two key metrics shown in Figure 1: (1) prediction loss dynamics and (2) attention score evolution, evaluated under flat ($\{0.1, 0.2, 0.4\}$) and sharp ($\{1.0, 1.5, 2.0\}$) curvature regimes. The prediction loss is computed as the average squared loss over prompts containing query token $v_k$. For attention scores, we track $v_1$'s self-attention score $\text{Attn}_1^{(t)}$ and other features' attention scores $\text{Attn}_{k,1}^{(t)}$ ($k \in \{2, 3, 4\}$) on $v_1$ at each epoch.

For flat $L$-regime, as shown in Figure 1(a) and Figure 1(c), we observe two distinct training phases. For example, with $L = 0.1$, the prediction loss rapidly decreases to around 1.0 by epoch $t = 40$, driven by increasing attention on the target feature shown in Figure 1(c). Subsequently, it steadily declines to near zero by $t = 250$. At the same time, $\text{Attn}_1^{(t)}$ approaches 1 under the transformer parameter $\theta$. The convergence time shortens with increasing $L$ due to stronger gradient updates, consistent with Theorem 1, Proposition 1, Proposition 2, and Proposition 3.

As depicted in Figure 1(b) and Figure 1(d), the sharp $L$-regime exhibits different three training phases. Specifically, when $L = 1$, the prediction loss drops rapidly to approximately $0.6$ by $t = 30$, then decreases steadily to around $0.1$ by $t = 110$. After that, it converges slowly toward $0$ by $t = 200$. The dynamics of the attention scores in Figure 1(d) exhibit the same three-phase progression. Additionally, under our experiment setting, the convergence upper bounds $T_f^*$ in Theorem 1 and $T_s^*$ in Theorem 2 satisfy $T_f^* > T_s^*$, indicating that the sharp regime achieves a faster convergence time. These empirical observations are consistent with our theoretical predictions in Theorem 2 and Proposition 4.

## 7 CONCLUSIONS AND LIMITATIONS

We presented provable results showing how transformers can learn a broad family of nonlinear tasks in context, identifying two distinct training regimes governed by task curvature. These findings illuminate the basic mechanisms-attention concentration and curvature-dependent gradient dynamics—that underpin ICL. By revealing how Lipschitz continuity and feature separation jointly determine convergence and generalization, our theory offers testable predictions for real-world settings (e.g., effects of task smoothness and feature geometry) and provides a principled starting point for extending formal guarantees to more realistic transformer architectures. Although our analysis focuses on a single-layer, single-head model, extending it to multi-head and multi-layer transformers presents substantial challenges due to intertwined cross-head gradients and the recursive evolution of representations across layers. Depth introduces residual connections and nonlinearities that couple representation learning with attention dynamics, making phase-wise analysis significantly more delicate. We view developing such extensions as an important direction for future work, and our curvature-sensitive framework provides a promising starting point.

## REPRODUCIBILITY STATEMENT

We are committed to ensuring the reproducibility of our work. All theoretical claims made in this paper are supported by detailed, step-by-step proofs, which can be found in Appendices B through G. The key assumptions underlying our analysis, including the non-degenerate L-Lipschitz function class and the one-layer transformer architecture, are formally defined in the System Model (Section 3), and we have added additional discussion in Section 7. For our empirical validation, the setup, hyperparameters, and task details for the main experiments are described in Section 6 and Appendix B. To facilitate direct replication of all figures and results, we have included the complete source code, which contains the model implementation, data generation, and training procedures, as part of the supplementary materials. Together, these materials satisfy the ICLR guidelines for reproducibility and allow others to reproduce our theoretical analyses and empirical findings.

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

# APPENDIX

In the appendices, we first present additional experimental results in Appendix B. Then we present detailed proofs of Lemma 1 and Lemma 2 in Appendix C and Appendix D. For ease of exposition, before establishing the two main theorems, we separately analyze the convergence dynamics under both the flat $L$-regime and the sharp $L$-regime.

Specifically, following the proof of Lemma 1, we prove Proposition 2 (see Appendix E.1) and Proposition 3 (see Appendix E.2) as preliminaries for the proof of Theorem 1 in the flat $L$-regime (see Appendix E.3).

We then proceed to prove Proposition 4 (see Appendix F.1) as a preliminary for Theorem 2 (see Appendix F.2) in the sharp $L$-regime.

Finally, based on Theorem 1 and Theorem 2, we prove Proposition 1 (see Appendix G).

## A    THE USE OF LLMs FOR POLISHING WRITING

We used a large language model (LLM) tool (e.g., ChatGPT) solely to polish and improve the readability of certain sections of this manuscript—specifically, the introduction, abstract, and several explanatory paragraphs. All research ideas, theoretical derivations, experimental designs, analyses, and conclusions are original to the authors. The LLM was not used to generate technical content, proofs, or results. We reviewed and edited all text produced with LLM assistance to ensure accuracy and consistency with our intended meaning.

## B    ADDITIONAL EXPERIMENTAL RESULTS

Building on the prediction loss and attention dynamics shown in Figure 1, we provide a more detailed analysis of the $4 \times 4$ attention maps in this appendix. Figure 2 and Figure 3 show the attention patterns for the flat ($L = 0.1$) and sharp ($L = 1$) loss landscapes, respectively, under the experimental conditions of Section 6.

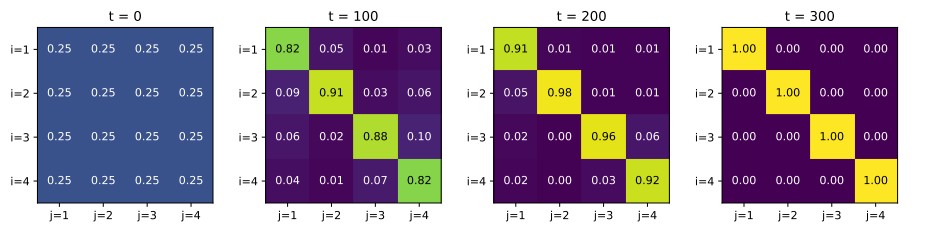

Figure 2: Dynamics of attention maps under flat $L = 0.1$

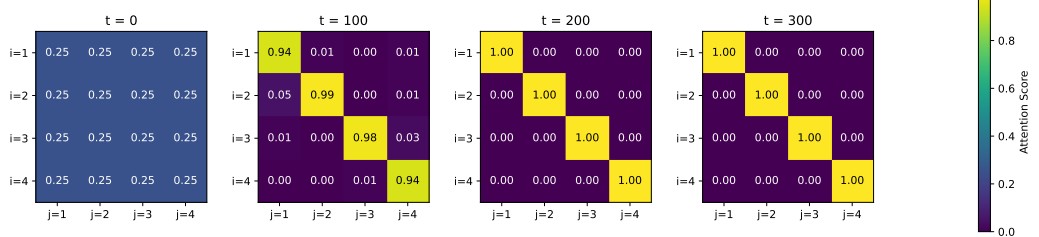

Figure 3: Dynamics of attention maps under sharp $L = 1$

In the attention maps, each grid position $(i, j)$ corresponds to the attention score $\text{Attn}_{i,j}$, where $i, j \in \{1, 2, 3, 4\}$ are the indices of key and query, respectively. An important property is that the sum of attention scores for each column equals one ($\sum_i \text{Attn}_{i,j} = 1$), given the same query $v_j$. The

attention maps show a clear trend: the diagonal entries $(i, i)$ become progressively darker over $t$, indicating that the corresponding self-attention scores $\text{Attn}_i$ increase for all $i \in \{1, 2, 3, 4\}$. We observe these scores approaching 1 before $t = 300$ for the flat landscape $L = 0.1$ and before $t = 200$ for the sharp landscape $L = 1$. In contrast, the off-diagonal entries $(i, j)$, where $i \neq j$, become lighter, with $\text{Attn}_{i,j}$ converging toward zero. This behavior supports the theoretical findings presented in Proposition 2 and Proposition 3 and is directly reflected in the attention score dynamics plotted in Figure 1(c) and Figure 1(d).

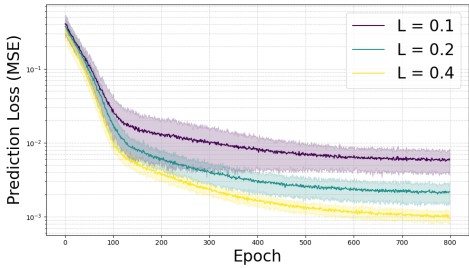 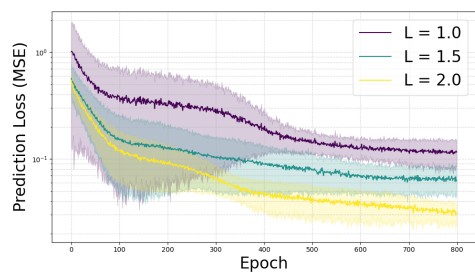

(a) Prediction loss for flat $L$-set under polynomial functions.

(b) Prediction loss for sharp $L$-set under polynomial functions.

Figure 4: Training dynamics of prediction losses for two sets of $L$: flat ($\{0.1, 0.2, 0.4\}$) and sharp ($\{1.0, 1.5, 2.0\}$) under polynomial functions.

To further validate the generality of our theoretical results, we conducted an additional experiment on a different class of nonlinear functions: random second-degree polynomials of the form $f(x) = x^\top A x + b^\top x$, where the matrix $A$ and vector $b$ were randomly generated for each task instance. The elements of $A$ and $b$ satisfy $A_{ij}, b_i \sim \mathcal{N}(0, (\frac{L}{d})^2)$. For this experiment, we relaxed the finite feature set assumption by sampling input features from a continuous Gaussian distribution. In other words, there can be infinitely many feature vectors. For the other parameters, we set $d = 30, N = 150, M = 500$. The prediction loss, averaged over 30 independent runs, is shown below for both the flat $L$-regime and the sharp $L$-regime. The results strongly corroborate our main findings: The flat regime has two training phases, while the sharp regime has three phases. Within both regimes, a larger Lipschitz constant $L$ consistently results in faster convergence of the prediction loss. This provides further evidence that the identified training dynamics and the role of the Lipschitz constant hold for a broader class of nonlinear functions beyond the trigonometric family, and also hold for more general feature set and token sampling process.

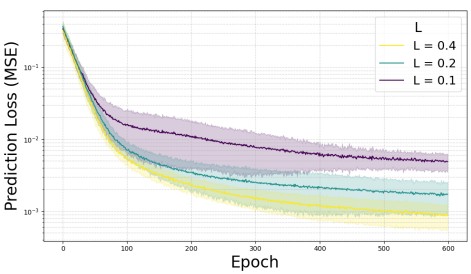 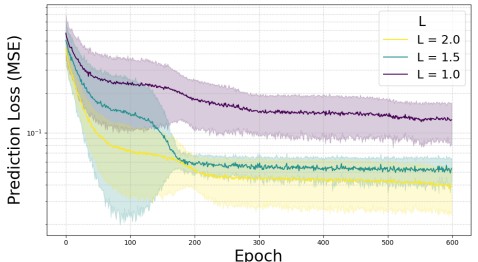

(a) Prediction loss for flat $L$-set under two-layer transformer.

(b) Prediction loss for sharp $L$-set under two-layer transformer.

Figure 5: Training dynamics of prediction losses for two sets of $L$: flat ($\{0.1, 0.2, 0.4\}$) and sharp ($\{1.0, 1.5, 2.0\}$) under two-layer transformer.

In addition to the single-layer setting analyzed in the main text, we further evaluate whether the curvature-dependent convergence behavior persists in deeper Transformer architectures such as two-layer and four-layer models. To this end, we present experiments using more realistic architectures that contain two or four stacked self-attention layers followed by two or four FFN blocks, under the same

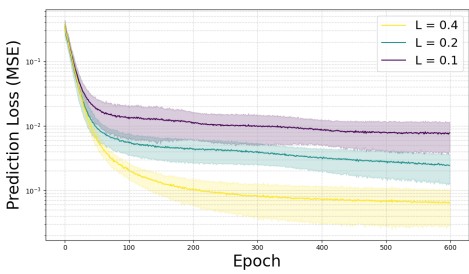 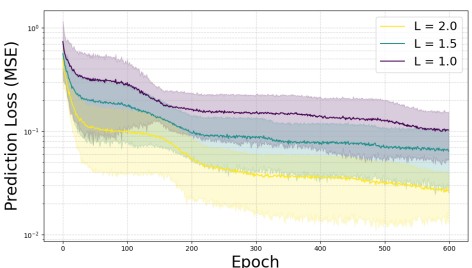

(a) Prediction loss for flat $L$-set under four-layer transformer.

(b) Prediction loss for sharp $L$-set under four-layer transformer.

Figure 6: Training dynamics of prediction losses for two sets of $L$: flat ($\{0.1, 0.2, 0.4\}$) and sharp ($\{1.0, 1.5, 2.0\}$) under four-layer transformer.

data and task setup as in Figure 4. As shown in Figure 5 and Figure 6, the deeper Transformers exhibit the same qualitative learning dynamics predicted by our theory, including a clear phase transition between the flat and sharp $L$-regimes and consistent phase-wise convergence behavior. These results indicate that the core mechanisms identified in our analysis, which were derived for a single-layer model for analytical tractability, naturally extend to multi-layer Transformer architectures.

## C  PROOF OF LEMMA 1

Recall the definition of the prediction error $\mathcal{L}(P; Q)$ in Eq. (5):

$$\mathcal{L}(P; Q) = \frac{1}{2} \sum_{k=1}^{K} \mathbb{E}\left[\mathbb{1}\{x_{\text{query}} = v_k\}(\hat{y}_{\text{query}} - f(v_k))^2\right]$$

$$= \frac{1}{2} \sum_{k=1}^{K} \mathbb{E}\left[\mathbb{1}\{x_{\text{query}} = v_k\}\left(\sum_{n\in[K]} \text{Attn}_n^{(t)} f(v_n) - f(v_k)\right)^2\right]$$

$$= \frac{1}{2} \sum_{k=1}^{K} \mathbb{E}\left[\mathbb{1}\{x_{\text{query}} = v_k\}\left(\sum_{n\in[K]} \text{Attn}_n^{(t)}(f(v_n) - f(v_k))\right)^2\right]$$

$$= \frac{1}{2} \sum_{k=1}^{K} \mathbb{E}\left[\mathbb{1}\{x_{\text{query}} = v_k\}\left(\sum_{n\neq k} \text{Attn}_n^{(t)}(f(v_n) - f(v_k))\right)^2\right]$$

where we apply $\sum_{n\in[K]} \text{Attn}_n^{(t)} = 1$ to rewrite $f(v_k)$ as $\sum_{n\in[K]} \text{Attn}_n^{(t)} f(v_k)$ in the third equality.

Next, under the non-degenerate $L$-Lipschitz condition of the function class (see Eq. (2)), we have $\sum_{n\neq k} |f(v_n) - f(v_k)| = \mathcal{O}(L\Delta)$. Therefore, the sum can be bounded (order-wise) as

$$\left|\sum_{n\neq k} \text{Attn}_n^{(t)}(f(v_n) - f(v_k))\right| = (1 - \text{Attn}_k^{(t)}) \cdot \mathcal{O}(L\Delta)$$

and thus

$$\left(\sum_{n\neq k} \text{Attn}_n^{(t)}(f(v_n) - f(v_k))\right)^2 = (1 - \text{Attn}_k^{(t)})^2 \cdot \mathcal{O}(L^2\Delta^2).$$

Putting these together, we obtain:

$$\mathcal{L}(P; Q) = \frac{1}{2} \sum_{k=1}^{K} \mathbb{E}\left[\mathbb{1}\{x_{\text{query}} = v_k\}(1 - \text{Attn}_k^{(t)})^2 \cdot \mathcal{O}(L^2\Delta^2)\right].$$

This completes the proof of Lemma 1.

## D    PROOF OF LEMMA 2

To derive $g_k$, we first compute the gradient of prediction loss with respect to $Q^{(t)}$ as follows:

$$\nabla_{Q^{(t)}}\mathcal{L} = \mathbb{E}\Big[(\hat{y}_{\text{query}} - f(v_k))^{\top} \frac{\partial \hat{y}_{\text{query}}}{\partial Q^{(t)}}\Big] = \mathbb{E}\Big[(\hat{y}_{\text{query}} - f(v_k))^{\top} \sum_{i \in [N]} \frac{\partial \text{attn}_i}{\partial Q^{(t)}} y_i\Big].$$

We then compute the gradient of $\text{attn}_i^{(t)}$ with respect to $Q^{(t)}$:

$$\frac{\partial \text{attn}_i^{(t)}}{\partial Q^{(t)}} = \frac{e^{\bar{X}_i^{\top} Q^{(t)} x_{\text{query}}} \cdot \bar{X}_i^{\top} x_{\text{query}}^{\top}(\sum_{j \in [N]} e^{\bar{X}_j^{\top} Q^{(t)} x_{\text{query}}})}{(\sum_{j \in [N]} e^{\bar{X}_j^{\top} Q^{(t)} x_{\text{query}}})^2}$$

$$- \frac{\sum_{j \in [N]} e^{\bar{X}_j^{\top} Q^{(t)} x_{\text{query}}} \cdot \bar{X}_j^{\top} x_{\text{query}}^{\top} \cdot e^{\bar{X}_i^{\top} Q^{(t)} x_{\text{query}}}}{(\sum_{j \in [N]} e^{\bar{X}_j^{\top} Q^{(t)} x_{\text{query}}})^2}$$

$$= \text{attn}_i^{(t)} \cdot \bar{X}_i^{\top} x_{\text{query}} - \text{attn}_i^{(t)} \sum_{j \in [N]} \text{attn}_j^{(t)} \cdot \bar{X}_j^{\top} x_{\text{query}}$$

$$= \text{attn}_i^{(t)} \sum_{j \in [N]} \text{attn}_j^{(t)} (\bar{X}_i - \bar{X}_j) x_{\text{query}}^{\top}.$$

Substituting into the above expression gives:

$$\nabla_{Q^{(t)}}\mathcal{L} = \mathbb{E}\Big[(\hat{y}_{\text{query}}^{(t)} - f(v_k)) \sum_{i,j \in [N]} \text{attn}_i^{(t)} \text{attn}_j^{(t)} (\bar{X}_i - \bar{X}_j) x_{\text{query}}^{\top} y_i\Big].$$

For any feature vectors $v_k$ and $v_{k'}$, we calculate

$$v_{k'}^{\top} \nabla_{Q^{(t)}}\mathcal{L} \cdot v_k = \mathbb{E}\Big[\mathbb{1}\{x_{\text{query}} = v_k\}(\hat{y}_{\text{query}} - f(v_k)) \sum_{i,j \in [N]} \text{attn}_i^{(t)} \text{attn}_j^{(t)} y_i v_{k'}^{\top}(\bar{X}_i - \bar{X}_j)\Big]$$

$$= \mathbb{E}\Big[\mathbb{1}\{x_{\text{query}} = v_k\}(\hat{y}_{\text{query}} - f(v_k)) \sum_{m,n \in [K]} \sum_{i \in \mathcal{V}_m} \sum_{j \in \mathcal{V}_n} \text{attn}_i^{(t)} \text{attn}_j^{(t)} y_i v_{k'}^{\top}(v_m - v_n)\Big]$$

$$= \mathbb{E}\Big[\mathbb{1}\{x_{\text{query}} = v_k\}(\hat{y}_{\text{query}} - f(v_k)) \sum_{n \in [K]} \sum_{i \in \mathcal{V}_{k'}} \sum_{j \in \mathcal{V}_n} \text{attn}_i^{(t)} \text{attn}_j^{(t)} y_i v_{k'}^{\top}(v_{k'} - v_n)\Big]$$

$$+ \mathbb{E}\Big[\mathbb{1}\{x_{\text{query}} = v_k\}(\hat{y}_{\text{query}} - f(v_k)) \sum_{m \in [K]} \sum_{i \in \mathcal{V}_m} \sum_{j \in \mathcal{V}_{k'}} \text{attn}_i^{(t)} \text{attn}_j^{(t)} y_i v_{k'}^{\top}(v_m - v_{k'})\Big]$$

$$= \mathbb{E}\Big[\mathbb{1}\{x_{\text{query}} = v_k\}(\hat{y}_{\text{query}} - f(v_k)) \text{Attn}_{k'}^{(t)} f(v_{k'}) \sum_{n \in [K]} \text{Attn}_n^{(t)}\Big]$$

$$- \mathbb{E}\Big[\mathbb{1}\{x_{\text{query}} = v_k\}(\hat{y}_{\text{query}} - f(v_k)) \text{Attn}_{k'}^{(t)} \sum_{m \in [K]} \text{Attn}_m^{(t)} f(v_m)\Big]$$

$$= \mathbb{E}\Big[\mathbb{1}\{x_{\text{query}} = v_k\}(\hat{y}_{\text{query}} - f(v_k)) \text{Attn}_{k'}^{(t)} \sum_{m \in [K]} \text{Attn}_m^{(t)} (f(v_{k'}) - f(v_m))\Big].$$

Since $\hat{y}_{\text{query}} = \sum_{i \in [N]} \text{attn}_i y_i = \sum_{m \in [K]} \text{Attn}_m^{(t)} f(v_m)$, we obtain

$$v_{k'}^{\top} \nabla_{Q^{(t)}}\mathcal{L} v_k = -\mathbb{E}\Big[\mathbb{1}\{x_{\text{query}} = v_k\} \text{Attn}_{k'}^{(t)} \sum_{n \in [K]} \sum_{m \in [K]} \text{Attn}_m^{(t)} \text{Attn}_n^{(t)} (f(v_k) - f(v_n))(f(v_{k'}) - f(v_m))\Big]$$

$$= -\mathbb{E}\Big[\mathbb{1}\{x_{\text{query}} = v_k\} \text{Attn}_{k'}^{(t)} (f(v_k) - \sum_{n \in [K]} \text{Attn}_n^{(t)} f(v_n))(f(v_{k'}) - \sum_{m \in [K]} \text{Attn}_m^{(t)} f(v_m))\Big].$$

We then derive Eq. (10) by letting $v_k = v_{k'}$ using this expression:

$$g_k^{(t)} = -v_k^{\top} \nabla_{Q^{(t)}}\mathcal{L} \cdot v_k$$

$$
= \mathbb{E}\left[\mathbb{1}\{x_{\text{query}} = v_k\} \operatorname{Attn}_k^{(t)}\left(f(v_k) - \sum_{n\in[K]} \operatorname{Attn}_n^{(t)} f(v_n)\right)^2\right]
$$

$$
= \mathbb{E}\left[\mathbb{1}\{x_{\text{query}} = v_k\} \operatorname{Attn}_k^{(t)}\left(\sum_{n\in[K]} \operatorname{Attn}_n^{(t)}(f(v_k) - f(v_n))\right)^2\right]
$$

$$
= \mathbb{E}\left[\mathbb{1}\{x_{\text{query}} = v_k\} \operatorname{Attn}_k^{(t)}(1 - \operatorname{Attn}_k^{(t)})^2\, \Theta(L^2\Delta^2)\right],
$$

where the last equality follows from the non-degenerate $L$-Lipschitz condition in Eq. (2).

For any $k \neq n$, we calculate

$$
|g_{k,k'}^{(t)}| = \mathbb{E}\left[\mathbb{1}\{x_{\text{query}} = v_k\}\operatorname{Attn}_{k'}^{(t)}|f(v_k) - \sum_{j\in[K]} \operatorname{Attn}_j^{(t)} f(v_j)| \cdot |f(v_{k'}) - \sum_{m\in[K]} \operatorname{Attn}_m^{(t)} f(v_m)|\right]
$$

$$
= p_k \cdot \mathbb{E}\left[\operatorname{Attn}_{k'}^{(t)}\Big|\sum_{j\in[K]} (\operatorname{Attn}_j^{(t)} f(v_k) - \operatorname{Attn}_j^{(t)} f(v_j))\Big| \cdot \Big|\sum_{m\in[K]} (\operatorname{Attn}_m^{(t)} f(v_{k'}) - \operatorname{Attn}_m^{(t)} f(v_m))\Big|\right]
$$

$$
= p_k \cdot \mathbb{E}\left[\operatorname{Attn}_{k'}^{(t)} \cdot \sum_{j\in[K]} \operatorname{Attn}_j^{(t)}|f(v_k) - f(v_j)| \cdot \sum_{m\in[K]} \operatorname{Attn}_m^{(t)}|f(v_{k'}) - f(v_m)|\right]
$$

$$
= p_k \cdot \mathbb{E}\left[\operatorname{Attn}_{k'}^{(t)} \cdot (1 - \operatorname{Attn}_k^{(t)}) \cdot (1 - \operatorname{Attn}_{k'}^{(t)}) \cdot \Theta(L^2\Delta^2)\right].
$$

This completes the proof of Lemma 2.

# E  PROOFS FOR FLAT $L$-REGIME

## E.1  PROOF OF PROPOSITION 2

**Lemma 3.** *For any $t \in \{1, \ldots, T_f^1\}$, if $x_{\text{query}} = v_k$, we have:*

- $\operatorname{Attn}_k^{(t)} = \Omega\left(\frac{1}{K}\right)$,

- $1 - \operatorname{Attn}_k^{(t)} = \Theta(1)$,

- $\operatorname{Attn}_n^{(t)} = \Theta\left(\frac{1 - \operatorname{Attn}_k^{(t)}}{K}\right) = \Theta\left(\frac{1}{K}\right)$ *for all $n \neq k$.*

*Proof.* Fix any $t \in \{1, \ldots, T_f^1\}$. By definition,

$$
\operatorname{Attn}_k^{(t)} = \frac{|\mathcal{V}_k|e^{v_k^\top Q^{(t)} v_k}}{\sum_{j\in[N]} e^{E_j^{x\top} Q^{(t)} v_k}} = \frac{|\mathcal{V}_k|e^{q_k^{(t)}}}{\sum_{m\neq k} |\mathcal{V}_m|e^{q_{k,m}^{(t)}} + |\mathcal{V}_k|e^{q_k^{(t)}}}
$$

$$
= \frac{1}{\sum_{m\neq k} \frac{|\mathcal{V}_m|}{|\mathcal{V}_k|}\exp(q_{k,m}^{(t)} - q_k^{(t)}) + 1}.
$$

By the symmetry property in the initial phase, $q_{k,m}^{(t)} = \Theta\left(\frac{q_k^{(t)}}{K}\right)$. Thus,

$$
e^{-(\log K + \Theta(\frac{\log K}{K}))} \leq \exp(q_{k,m}^{(t)} - q_k^{(t)}) \leq e^{\Theta(\frac{\log K}{K})}.
$$

Define $u_k = K(p_k - \delta)$ and $U_k = K(p_k + \delta)$. Then, from the concentration property (see Eq. (7)), $|\mathcal{V}_k| \in [\frac{u_k}{K}N, \frac{U_k}{K}N]$ for constants $u_k, U_k = \Theta(1)$. Therefore,

$$
\operatorname{Attn}_k^{(t)} \geq \frac{1}{e^{\Theta(\frac{\log K}{K})}(\frac{N}{|\mathcal{V}_k|} - 1) + 1} \geq \frac{1}{e^{\Theta(\frac{\log K}{K})}(\frac{K}{u_k} - 1) + 1} = \Omega\left(\frac{1}{K}\right).
$$

For the upper bound,

$$
\operatorname{Attn}_k^{(t)} \leq \frac{1}{e^{-(\log K + \Theta(\frac{\log K}{K}))}(\frac{N}{|\mathcal{V}_k|} - 1) + 1} \leq \frac{1}{e^{-1}(\frac{1}{U_k} - \frac{1}{K}) + 1},
$$

which follows because $U_k = \Theta(1)$, and hence

$$1 - \text{Attn}_k^{(t)} \geq \frac{\frac{1}{U_k} - \frac{1}{K}}{(\frac{1}{U_k} - \frac{1}{K}) + e} = \Theta(1).$$

The reverse bound is similar, showing $1 - \text{Attn}_k^{(t)} = \Theta(1)$.

For $n \neq k$, by similar calculation,

$$\text{Attn}_n^{(t)} = \frac{|\mathcal{V}_n| \exp(q_{k,n}^{(t)})}{\sum_{m \neq k} |\mathcal{V}_m| \exp(q_{k,m}^{(t)}) + |\mathcal{V}_k| \exp(q_k^{(t)})},$$

and since $|\mathcal{V}_m|/|\mathcal{V}_n| = \Theta(1)$ and $\exp(q_{k,m}^{(t)} - q_{k,n}^{(t)}) = e^{O(\frac{\log K}{K})}$,

$$\frac{\text{Attn}_n^{(t)}}{1 - \text{Attn}_k^{(t)}} = \frac{|\mathcal{V}_n| \exp(q_{k,n}^{(t)})}{\sum_{m \neq k} |\mathcal{V}_m| \exp(q_{k,m}^{(t)})} = \Theta\left(\frac{1}{K}\right).$$

Thus, $\text{Attn}_n^{(t)} = (1 - \text{Attn}_k^{(t)})\Theta\left(\frac{1}{K}\right) = \Theta\left(\frac{1}{K}\right)$, since $1 - \text{Attn}_k^{(t)} = \Theta(1)$. $\qquad\square$

**Lemma 4.** *For any $t \in \{1, \ldots, T_f^1\}$, given $x_{\text{query}} = v_k$, we have:*

- $g_k^{(t)} = \Theta\left(\frac{L^2\Delta^2}{K}\right)$,

- $|g_{k,n}^{(t)}| = O\left(\frac{L^2\Delta^2}{K^2}\right)$ *for any $n \neq k$.*

*Proof.* By the gradient expression from Lemma 2, we have

$$g_k^{(t)} = \mathbb{E}\left[\mathbb{1}\{x_{\text{query}} = v_k\} \text{Attn}_k^{(t)}(1 - \text{Attn}_k^{(t)})^2 \Theta(L^2\Delta^2)\right]$$

$$= p_k \cdot \mathbb{E}\left[\text{Attn}_k^{(t)}(1 - \text{Attn}_k^{(t)})^2 \mid x_{\text{query}} = v_k\right] \cdot \Theta(L^2\Delta^2).$$

By Lemma 3, in Phase I, we have

$$p_k = \Theta(1/K), \quad \text{Attn}_k^{(t)} = \Theta(1), \quad \text{and} \quad 1 - \text{Attn}_k^{(t)} = \Theta(1).$$

Therefore,

$$g_k^{(t)} = \Theta\left(\frac{1}{K} \cdot 1 \cdot (1)^2 \cdot L^2\Delta^2\right) = \Theta\left(\frac{L^2\Delta^2}{K}\right).$$

For the cross-gradient term, for $n \neq k$,

$$|g_{k,n}^{(t)}| = \mathbb{E}\left[\mathbb{1}\{x_{\text{query}} = v_k\} \text{Attn}_n^{(t)} |f(v_k) - \sum_j \text{Attn}_j^{(t)} f(v_j)| \, |f(v_n) - \sum_m \text{Attn}_m^{(t)} f(v_m)|\right]$$

$$= p_k \cdot \mathbb{E}\left[\text{Attn}_n^{(t)} \left| \sum_j \text{Attn}_j^{(t)}(f(v_k) - f(v_j)) \right| \left| \sum_m \text{Attn}_m^{(t)}(f(v_n) - f(v_m)) \right| \mid x_{\text{query}} = v_k\right]$$

$$\leq p_k \cdot \mathbb{E}\left[\text{Attn}_n^{(t)} \cdot \sum_j \text{Attn}_j^{(t)} |f(v_k) - f(v_j)| \cdot \sum_m \text{Attn}_m^{(t)} |f(v_n) - f(v_m)|\right]$$

$$= p_k \cdot \mathbb{E}\left[\text{Attn}_n^{(t)}(1 - \text{Attn}_k^{(t)})(1 - \text{Attn}_n^{(t)})\mathcal{O}(L^2\Delta^2)\right].$$

By Lemma 3, $\text{Attn}_n^{(t)} = \Theta(1/K)$ and $1 - \text{Attn}_k^{(t)}, 1 - \text{Attn}_n^{(t)} = \Theta(1)$, so

$$|g_{k,n}^{(t)}| = \mathcal{O}\left(\frac{1}{K} \cdot \frac{1}{K} \cdot 1 \cdot 1 \cdot L^2\Delta^2\right) = \mathcal{O}\left(\frac{L^2\Delta^2}{K^2}\right).$$

This completes the proof of Lemma 4. $\qquad\square$

**Lemma 5.** *Given $\delta = o(1)$, at the end of Phase I (i.e., at $t = T_f^1 + 1$), we have:*

- $q_k^{(T_f^1+1)} = \Theta(\log K)$,

- $\text{Attn}_k^{(T_f^1+1)} = \Omega\left(\frac{1}{1+\delta}\right)$ *if $x_{\text{query}} = v_k$.*

*Proof.* By Lemma 4, we have $g_k^{(t)} = \Theta\left(\frac{L^2\Delta^2}{K}\right)$ for all $t$ in Phase I. Thus,

$$q_k^{(T_f^1+1)} = q_k^{(0)} + \eta \sum_{t=1}^{T_f^1} g_k^{(t)}$$

$$= q_k^{(0)} + \eta \cdot T_f^1 \cdot \Theta\left(\frac{L^2\Delta^2}{K}\right)$$

$$= \Theta(\log K),$$

where we apply the definition of $T_f^1$ from the main text.

For the off-diagonal terms, from Lemma 4, $|g_{k,m}^{(t)}| = O\left(\frac{L^2\Delta^2}{K^2}\right)$ for any $m \neq k$. Hence,

$$q_{k,m}^{(T_f^1+1)} \leq |q_{k,m}^{(0)}| + \eta \cdot T_f^1 \cdot O\left(\frac{L^2\Delta^2}{K^2}\right)$$

$$= \mathcal{O}\left(\frac{\log K}{K}\right).$$

Therefore, at the end of Phase I,

$$q_k^{(T_f^1+1)} - q_{k,m}^{(T_f^1+1)} = \Theta(\log K) - \mathcal{O}\left(\frac{\log K}{K}\right) = \Theta(\log K).$$

Now, the attention weight for $k$ at time $t$ is

$$\text{Attn}_k^{(t)} = \frac{1}{\sum_{m \neq k} \frac{|\mathcal{V}_m|}{|\mathcal{V}_k|} \exp(q_{k,m}^{(t)} - q_k^{(t)}) + 1}.$$

Using the above, for $t = T_f^1 + 1$,

$$\exp(q_{k,m}^{(t)} - q_k^{(t)}) \leq \exp\left(\mathcal{O}\left(\frac{\log K}{K}\right) - \log K\right) = \mathcal{O}\left(\frac{1}{K}\right).$$

By the concentration condition in Eq. (7), $|\mathcal{V}_k| \geq \frac{u_k}{K}N$ for some $u_k = \Theta(1)$, and $N/|\mathcal{V}_k| = \Theta(1/\delta)$ (since $\delta = o(1)$ is the imbalance parameter). Thus,

$$\text{Attn}_k^{(t)} \geq \frac{1}{\mathcal{O}\left(\frac{1}{K}\right)\left(\frac{N}{|\mathcal{V}_k|} - 1\right) + 1} \geq \frac{1}{\mathcal{O}\left(\frac{1}{u_k} - \frac{1}{K}\right) + 1} = \Omega\left(\frac{1}{1+\delta}\right),$$

where the last equality follows because $1/u_k - 1/K = \Theta(\delta)$ from Eq. (7).

This completes the proof of Lemma 5 and Proposition 2. $\qquad\square$

### E.2 PROOF OF PROPOSITION 3

**Lemma 6.** *For any $t \in \{T_f^1 + 1, \ldots, T_f^*\}$, given $\delta = o(1)$, if $x_{\text{query}} = v_k$, we have:*

- $\text{Attn}_k^{(t)} = \Omega\left(\frac{1}{1+\delta}\right),$

- $1 - \text{Attn}_k^{(t)} = \mathcal{O}(\delta),$

- $\text{Attn}_n^{(t)} = \Theta\left(\frac{1-\text{Attn}_k^{(t)}}{K}\right) = \Theta\left(\frac{\delta}{K}\right)$ *for any $n \neq k$.*

*Proof.* By Proposition 2 (see also Lemma 5), for any $t \geq T_f^1 + 1$, we have $\text{Attn}_k^{(t)} = \Omega\left(\frac{1}{1+\delta}\right)$.

We now show that $1 - \text{Attn}_k^{(t)} = \mathcal{O}(\delta)$. Using the same attention formula as before,

$$\text{Attn}_k^{(t)} = \frac{1}{\sum_{m \neq k} \frac{|\mathcal{V}_m|}{|\mathcal{V}_k|} \exp(q_{k,m}^{(t)} - q_k^{(t)}) + 1}.$$

From previous bounds, $\exp(q_{k,m}^{(t)} - q_k^{(t)}) = O\left(\frac{1}{K}\right)$, and $|\mathcal{V}_k| \geq u_k N/K$ with $1/u_k - 1/K = \Theta(\delta)$. Therefore,

$$\text{Attn}_k^{(t)} \geq \frac{1}{\mathcal{O}\left(\frac{1}{K}\right)\left(\frac{N}{|\mathcal{V}_k|} - 1\right) + 1} \geq \frac{1}{\mathcal{O}\left(\frac{1}{u_k} - \frac{1}{K}\right) + 1} = \Omega\left(\frac{1}{1+\delta}\right).$$

For the upper bound, we compute:

$$1 - \text{Attn}_k^{(t)} \leq 1 - \frac{1}{\mathcal{O}\left(\frac{1}{K}\right)\left(\frac{N}{|\mathcal{V}_k|} - 1\right) + 1} = \frac{\mathcal{O}\left(\frac{1}{u_k} - \frac{1}{K}\right)}{\mathcal{O}\left(\frac{1}{u_k} - \frac{1}{K}\right) + 1} = \mathcal{O}(\delta),$$

where the last equality uses $\frac{1}{u_k} - \frac{1}{K} = \Theta(\delta)$ from Eq. (7). Thus, $1 - \text{Attn}_k^{(t)} = \mathcal{O}(\delta)$.

Finally, for any $n \neq k$, we can use the same method as in Lemma 3 to obtain:

$$\text{Attn}_n^{(t)} = \mathcal{O}\left(\frac{1 - \text{Attn}_k^{(t)}}{K}\right) = \mathcal{O}\left(\frac{\delta}{K}\right).$$

This completes the proof. $\qquad\square$

**Lemma 7.** *For any $t \in \{T_f^1 + 1, \ldots, T_f^*\}$ and any fixed $k \in [K]$, we have:*

- $g_k^{(t)} = \Theta\left(\frac{\delta^2 L^2 \Delta^2}{K}\right)$,

- $|g_{k,n}^{(t)}| = \mathcal{O}\left(\frac{\delta^2 L^2 \Delta^2}{K^2}\right)$ *for all $n \neq k$.*

*Proof.* Recall from the gradient expression and Lemma 4:

$$g_k^{(t)} = \mathbb{E}\left[\mathbb{1}\{x_{\text{query}} = v_k\} \text{Attn}_k^{(t)}(1 - \text{Attn}_k^{(t)})^2 \Theta(L^2\Delta^2)\right]$$

$$= p_k \cdot \mathbb{E}\left[\text{Attn}_k^{(t)}(1 - \text{Attn}_k^{(t)})^2 \mid x_{\text{query}} = v_k\right] \cdot \Theta(L^2\Delta^2).$$

By Lemma 5 and subsequent results for Phase II, we have $p_k = \Theta(1/K)$, $\text{Attn}_k^{(t)} = \Theta(1)$, and $1 - \text{Attn}_k^{(t)} = \Theta(\delta)$. Therefore,

$$g_k^{(t)} = \Theta\left(\frac{1}{K} \cdot 1 \cdot \delta^2 \cdot L^2\Delta^2\right) = \Theta\left(\frac{\delta^2 L^2 \Delta^2}{K}\right).$$

For the cross-gradient terms with $n \neq k$, using the same approach as in Lemma 4, we obtain

$$|g_{k,n}^{(t)}| \leq p_k \cdot \mathbb{E}\left[\text{Attn}_n^{(t)}(1 - \text{Attn}_k^{(t)})(1 - \text{Attn}_n^{(t)}) \cdot \mathcal{O}(L^2\Delta^2)\right].$$

In Phase II, by the previous lemma, we have $\text{Attn}_n^{(t)} = \Theta(\delta/K)$, $1 - \text{Attn}_n^{(t)} = \Theta(1)$, and $1 - \text{Attn}_k^{(t)} = \Theta(\delta)$. Thus,

$$|g_{k,n}^{(t)}| = \mathcal{O}\left(\frac{1}{K} \cdot \frac{\delta}{K} \cdot \delta \cdot 1 \cdot L^2\Delta^2\right) = \mathcal{O}\left(\frac{\delta^2 L^2 \Delta^2}{K^2}\right).$$

This completes the proof of Lemma 7. $\qquad\square$

**Lemma 8.** *At the end of Phase II under the flat L-regime (i.e., $t = T_f^* + 1$), if $x_{\text{query}} = v_k$, we have:*

- $q_k^{(T_f^*+1)} = \Theta\left(\frac{\log K}{\epsilon}\right)$,

- $\text{Attn}_k^{(T_f^*+1)} = \Omega\left(\frac{1}{1+\epsilon\delta}\right)$,

- $1 - \text{Attn}_k^{(T_f^*+1)} = \mathcal{O}(\epsilon\delta)$.

*Proof.* By Lemma 7, we have $g_k^{(t)} = \Theta\left(\frac{\delta^2 L^2 \Delta^2}{K}\right)$ in Phase II. Thus,

$$q_k^{(T_f^*+1)} = q_k^{(T_f^1)} + \eta \cdot \Theta\left(\frac{L^2\Delta^2\delta^2}{K}\right) \cdot (T_f^* - T_f^1)$$

$$= \Theta(\log(K\epsilon^{-1}))$$

$$= \Theta\left(\frac{\log K}{\epsilon}\right),$$

where the last step applies the scaling of $T_f^*$ and the learning rate in the flat $L$-regime.

For the cross terms, by Lemma 7 again,

$$q_{k,m}^{(T_f^*+1)} \leq |q_{k,m}^{(T_f^1)}| + \eta \cdot \mathcal{O}\left(\frac{\delta^2 L^2 \Delta^2}{K^2}\right) \cdot (T_f^* - T_f^1)$$

$$= \Theta\left(\frac{\log(K\epsilon^{-1})}{K}\right).$$

Therefore, at $t = T_f^* + 1$, we have

$$q_{k,m}^{(T_f^*+1)} - q_k^{(T_f^*+1)} = \mathcal{O}\left(\frac{\log(K\epsilon^{-1})}{K}\right) - \Theta(\log(K\epsilon^{-1})) = -\Theta(\log(K\epsilon^{-1})),$$

and so

$$\exp(q_{k,m}^{(t)} - q_k^{(t)}) \leq \exp\left(-\Theta(\log K)\right) = \mathcal{O}\left(\frac{1}{K}\right).$$

The attention weight for $k$ is then

$$\text{Attn}_k^{(t)} = \frac{1}{\sum_{m \neq k} \frac{|\mathcal{V}_m|}{|\mathcal{V}_k|} \exp(q_{k,m}^{(t)} - q_k^{(t)}) + 1}.$$

Using the bounds above, and $|\mathcal{V}_k| \geq u_k N/K$, with $\frac{1}{u_k} - \frac{1}{K} = \Theta(\delta)$ (see Eq. (7)), we obtain:

$$\text{Attn}_k^{(T_f^*+1)} \geq \frac{1}{\mathcal{O}(\epsilon) \cdot \mathcal{O}\left(\frac{1}{u_k} - \frac{1}{K}\right) + 1} = \Omega\left(\frac{1}{1 + \epsilon\delta}\right).$$

Similarly,

$$1 - \text{Attn}_k^{(T_f^*+1)} \leq \frac{\mathcal{O}(\epsilon) \cdot \mathcal{O}\left(\frac{1}{u_k} - \frac{1}{K}\right)}{\mathcal{O}(\epsilon) \cdot \mathcal{O}\left(\frac{1}{u_k} - \frac{1}{K}\right) + 1} = \mathcal{O}(\epsilon\delta).$$

This completes the proof of Lemma 8 and Proposition 3. $\qquad \square$

### E.3 PROOF OF THEOREM 1

Recall from Lemma 2 and its proof that the prediction error $\mathcal{L}(P; Q)$ defined in Eq. (5) can be expressed as

$$\mathcal{L}(P; Q) = \frac{1}{2} \sum_{k=1}^{K} \mathbb{E}\left[\mathbb{1}\{x_{\text{query}} = v_k\}(1 - \text{Attn}_k^{(t)})^2 \mathcal{O}(L^2\Delta^2)\right],$$

where we use $\sum_{n \neq k} \text{Attn}_n^{(t)} = 1 - \text{Attn}_k^{(t)}$ and, by the function class assumption, $|f(v_n) - f(v_k)| = \Theta(L\Delta)$.

At the end of Phase II (i.e., at $t = T_f^* + 1$), suppose $x_{\text{query}} = v_k$. By Lemma 8, we have $1 - \text{Attn}_k^{(T_f^*+1)} = O(\epsilon\delta)$. Therefore,

$$\mathcal{L}(P; Q) = \frac{1}{2} \sum_{k=1}^{K} \mathbb{E}\left[\mathbb{1}\{x_{\text{query}} = v_k\}(1 - \text{Attn}_k^{(T_f^*+1)})^2 \mathcal{O}(L^2\Delta^2)\right]$$

$$= \mathbb{E}\left[(1 - \text{Attn}_k^{(T_f^*+1)})^2 \mathcal{O}(L^2\Delta^2)\right]$$

$$= \mathcal{O}(\epsilon^2),$$

where the last equality uses $(1 - \text{Attn}_k^{(T_f^*+1)})^2 = \mathcal{O}(\epsilon^2\delta^2)$ and $L^2\Delta^2 = \mathcal{O}(1/(\Delta^2\delta^2)) \cdot \Delta^2 = \mathcal{O}(1/\delta^2)$ when $L \leq \Theta(1/(\Delta\delta))$, so the $\delta^2$ cancels, leaving $\mathcal{O}(\epsilon^2)$.

This establishes the desired rate and completes the proof of Theorem 1.

# F  PROOFS FOR SHARP $L$-REGIME

## F.1  PROOF OF PROPOSITION 4

**Lemma 9.** *For any $t \in \{T_f^* + 1, \ldots, T_s^*\}$ and any fixed $k \in [K]$, we have:*

- $g_k^{(t)} = \Theta\left(\frac{\delta^2 L^2 \Delta^2 \epsilon}{K}\right)$,

- $|g_{k,n}^{(t)}| = \mathcal{O}\left(\frac{\delta^2 L^2 \Delta^2 \epsilon}{K^2}\right)$ *for all $n \neq k$.*

*Proof.* Recall from the gradient expression:
$$g_k^{(t)} = \mathbb{E}\left[\mathbb{1}\{x_{\text{query}} = v_k\} \, \text{Attn}_k^{(t)} \, (1 - \text{Attn}_k^{(t)})^2 \, \Theta(L^2 \Delta^2)\right]$$
$$= p_k \cdot \mathbb{E}\left[\text{Attn}_k^{(t)}(1 - \text{Attn}_k^{(t)})^2 \mid x_{\text{query}} = v_k\right] \cdot \Theta(L^2 \Delta^2),$$
where $p_k = \Theta(1/K)$.

By Lemma 8, in this phase $\text{Attn}_k^{(t)} = \Theta(1)$ and $1 - \text{Attn}_k^{(t)} = \mathcal{O}(\epsilon\delta)$. Therefore,
$$g_k^{(t)} = \Theta\left(\frac{1}{K} \cdot 1 \cdot (\epsilon\delta)^2 \cdot L^2 \Delta^2\right) = \Theta\left(\frac{\delta^2 L^2 \Delta^2 \epsilon^2}{K}\right).$$

For the cross-gradient terms ($n \neq k$), by the same argument as in Lemma 7, we have:
$$|g_{k,n}^{(t)}| \leq p_k \cdot \mathbb{E}\left[\text{Attn}_n^{(t)}(1 - \text{Attn}_k^{(t)})(1 - \text{Attn}_n^{(t)}) \cdot \Theta(L^2 \Delta^2)\right].$$
In this phase, $\text{Attn}_n^{(t)} = \Theta(\delta/K)$, $1 - \text{Attn}_n^{(t)} = \Theta(1)$, and $1 - \text{Attn}_k^{(t)} = \mathcal{O}(\epsilon\delta)$. Therefore,
$$|g_{k,n}^{(t)}| \leq \Theta\left(\frac{1}{K} \cdot \frac{\delta}{K} \cdot \epsilon\delta \cdot 1 \cdot L^2 \Delta^2\right) = \mathcal{O}\left(\frac{\delta^2 L^2 \Delta^2 \epsilon}{K^2}\right).$$

This completes the proof of Lemma 9. $\qquad\square$

**Lemma 10.** *At the end of Phase II under the sharp $L$-regime (i.e., at $t = T_s^*$), if $x_{\text{query}} = v_k$, we have:*

- $q_k^{(T_s^*)} = \Theta\left(\frac{\log(KL\Delta)}{\epsilon}\right)$,

- $\text{Attn}_k^{(T_s^*)} = \Omega\left(\frac{1}{1+\epsilon\delta}\right)$,

- $1 - \text{Attn}_k^{(T_s^*)} = \mathcal{O}(\epsilon\delta)$.

*Proof.* By Lemma 9, for $t \in \{T_f^* + 1, \ldots, T_s^*\}$, we have $g_k^{(t)} = \Theta\left(\frac{\delta^2 L^2 \Delta^2 \epsilon}{K}\right)$. Thus,
$$q_k^{(T_s^*)} = q_k^{(T_f^*)} + \eta \cdot \Theta\left(\frac{L^2 \Delta^2 \delta^2 \epsilon}{K}\right) \cdot (T_s^* - T_f^*)$$
$$= \Theta(\log(KL\Delta\epsilon^{-1})),$$
using the total number of updates and the scaling of $T_s^*$. (Here, $T_s^* - T_f^* = \Theta\left(\frac{K \log(KL\Delta\epsilon^{-1})}{L^2 \Delta^2 \delta^2 \epsilon}\right)$.)

Similarly, for the cross-terms, by Lemma 9, $|g_{k,m}^{(t)}| = \mathcal{O}\left(\frac{\delta^2 L^2 \Delta^2 \epsilon}{K^2}\right)$ for $m \neq k$, and hence
$$q_{k,m}^{(T_s^*)} \leq |q_{k,m}^{(T_f^*)}| + \eta \cdot \mathcal{O}\left(\frac{\delta^2 L^2 \Delta^2 \epsilon}{K^2}\right) \cdot (T_s^* - T_f^*)$$
$$= \Theta\left(\frac{\log(KL\Delta\epsilon^{-1})}{K}\right).$$

Therefore,
$$q_{k,m}^{(T_s^*)} - q_k^{(T_s^*)} = -\Theta(\log(KL\Delta\epsilon^{-1})),$$

and so

$$\exp(q_{k,m}^{(T_s^*)} - q_k^{(T_s^*)}) = \mathcal{O}\left(\frac{\epsilon}{K}\right),$$

where the scaling in the sharp regime produces the $\epsilon$ factor.

For the attention, using the property from Lemma 3,

$$\text{Attn}_k^{(T_s^*)} = \frac{1}{\sum_{m \neq k} \frac{|\mathcal{V}_m|}{|\mathcal{V}_k|} \exp(q_{k,m}^{(T_s^*)} - q_k^{(T_s^*)}) + 1}.$$

By the previous bounds, and using $|\mathcal{V}_k| \geq u_k N/K$ and $\frac{1}{u_k} - \frac{1}{K} = \Theta(\delta)$, we obtain:

$$\text{Attn}_k^{(T_s^*)} \geq \frac{1}{\mathcal{O}(\epsilon) \cdot \mathcal{O}(\frac{1}{u_k} - \frac{1}{K}) + 1} = \Omega\left(\frac{1}{1 + \epsilon\delta}\right).$$

Finally,

$$1 - \text{Attn}_k^{(T_s^*)} \leq \frac{\mathcal{O}(\epsilon) \cdot \mathcal{O}(\frac{1}{u_k} - \frac{1}{K})}{\mathcal{O}(\epsilon) \cdot \mathcal{O}(\frac{1}{u_k} - \frac{1}{K}) + 1} = \mathcal{O}(\epsilon\delta),$$

which follows because $\frac{1}{u_k} - \frac{1}{K} = \Theta(\delta)$.

This completes the proof of Lemma 10 and Proposition 4. $\square$

### F.2 Proof of Theorem 2

As in the proof of Theorem 1, the prediction error $\mathcal{L}(P;Q)$ (from Eq. (5)) can be written as

$$\mathcal{L}(P;Q) = \frac{1}{2} \sum_{k=1}^{K} \mathbb{E}\left[\mathbb{1}\{x_{\text{query}} = v_k\}(1 - \text{Attn}_k^{(t)})^2 \mathcal{O}(L^2\Delta^2)\right].$$

Suppose $x_{\text{query}} = v_k$ at time $t = T_s^*$. By Lemma 10, we have $1 - \text{Attn}_k^{(T_s^*)} = O(\epsilon\delta)$. Therefore,

$$\mathcal{L}^{(T_s^*)}(P;Q) = \frac{1}{2} \sum_{k=1}^{K} \mathbb{E}\left[\mathbb{1}\{x_{\text{query}} = v_k\}(1 - \text{Attn}_k^{(T_s^*)})^2 \mathcal{O}(L^2\Delta^2)\right]$$

$$= \mathbb{E}\left[(1 - \text{Attn}_k^{(T_s^*)})^2 \mathcal{O}(L^2\Delta^2)\right]$$

$$= \mathcal{O}(\epsilon^2\delta^2).$$

Under the scaling regime for sharp $L$, either $\delta = o(1)$ or $\epsilon = o(1)$ (since the in-context learning regime assumes both go to zero), and hence $\mathcal{L}^{(T_s^*)}(P;Q) = \mathcal{O}(\epsilon^2)$ as required.

This completes the proof of Theorem 2.

## G Proof of Proposition 1

The result that $1 - \text{Attn}_k^{(t)} = \mathcal{O}(\epsilon)$ holds under both the flat $L$ regime and the sharp $L$ regime, as established in Lemma 8 and Lemma 10.

