# OpenReview forum: "Provable In-Context Learning of Nonlinear Regression with Transformers"
_ICLR.cc/2026/Conference — Submitted to ICLR 2026_

### Official Review · Reviewer_b5AQ · 2025-10-26

**Soundness:** 3
**Presentation:** 2
**Contribution:** 2
**Rating:** 4
**Confidence:** 4

**Summary:**

This paper studies the convergence of nonlinear Transformers on nonlinear regression tasks using ICL. The authors consider non-degenerate L-Lipschitz task functions and establish the analysis on flat and sharp curvature regimes, respectively. The results show that the learning phases and rates are different in the two cases based on the Lipschitz constant.

**Strengths:**

1. The problem of nonlinear regression with nonlinear Transformers using ICL is important and interesting.

2. The theoretical analysis is solid and impressive.

**Weaknesses:**

1. The contribution is not significant enough. The key reason is that the studied feature embedding is too simple by considering well-separated data with tiny or even no noise. This makes the mechanism of self-attention be to find context examples that share the same feature as the query. Such a mechanism is already discovered in previous works. Extending this result to nonlinear regression is not impressive enough since I believe this can be proved given well-separated data. A more challenging and interesting problem for nonlinear regression is what if there is no context input that is very close to the query.

2. The experiment results seem not aligned with the theory. The bound of $T\_s^\*$ in Theorem 2 shall be larger than $T_f^\*$ in Theorem 1. The authors also claim that "flat regime may enjoy faster convergence" in lines 305-306. However, the convergence of sharp regime in Figure 1(b) is faster than that of flat regime in Figure 1(a).

3. I am not sure how the results are related with the existing linear case analysis. Can the results be redueced to the linear case? Here is a key question, i.e., for the linear case, $L$ could be any arbitrary value $<\infty$. However, it seems that previous works do not divide the discussion into two cases like yours. Is it because your analysis is more fine-grained than theirs or anything else? I believe there should be some discussions here.

**Questions:**

1. In Eqn. 4, there is no $P$ in the right-hand side. Why not delete $P$?

2. Can you theoretically study the case where (1) $L$ is much larger than the case studied in Theorem 2, and/or (2) the noise is not $o(1)$ in the feature, and/or (3) $p_k$ is not uniform?

3. Are there any practical insights from the paper?

---

> ### Author Response · Authors · 2025-11-26
> **Response to Reviewer b5AQ (1/3)**
>
> Thank you for your thorough reviews and constructive comments. We provide our responses to your comments below and have made major revisions in our revised manuscript. To enhance clarity, we have highlighted the revised text in blue for easy identification.
>
> Q1. The contribution is not significant enough. The key reason is that the studied feature embedding is too simple by considering well-separated data with tiny or even no noise. This makes the mechanism of self-attention be to find context examples that share the same feature as the query. Such a mechanism is already discovered in previous works. Extending this result to nonlinear regression is not impressive enough since I believe this can be proved given well-separated data. A more challenging and interesting problem for nonlinear regression is what if there is no context input that is very close to the query.
>
> **A1**: We thank the reviewer for raising this concern and welcome the opportunity to clarify our contribution. Below we first clarify why our study is non-trivial, and then connect our results to the setting suggested by the reviewer.
>
> ** Why our study is non-trivial**
>
> We clarify that our learning problem is substantially more complex than simply identifying the nearest feature vector, even given well-separated clusters of features. In fact, the model must learn how the curvature of a class of $L$-Lipschitz functions interacts with the feature geometry, rather than merely retrieve a matching feature. This makes the analysis challenging and nontrivial, necessitating several new techniques that we introduce below.
>
> Novel contributions of this work
>
> - **Loss Decomposition via Attention and Feature Gaps:** We derive a new decomposition that explicitly relates the prediction loss to attention weights and cross-feature gaps under nonlinear functions (Lemma 1).
>
> - **Curvature-Governed Gradient Dynamics:** We characterize how **function curvature** (captured by the **Lipschitz constant**) determines the magnitude and evolution of attention gradients (Lemmas 2, 4, and 7 in Appendices E.1 and E.2). This leads to two distinct training regimes, i.e., flat vs. sharp curvature, with provably different convergence behaviors (Propositions 2-4), revealing a phase transition in attention dynamics.
>
> - **Curvature-Sensitive Convergence Guarantees:** Our convergence rates depend explicitly on the Lipschitz constant $L$, yielding function-dependent training times rather than the curvature-free behavior seen in linear settings (Theorems 1–2).
> This explicit and curvature-aware tracking of attention evolution forms the core novelty of our work, which are valuable to the community.
>
> **How our results connect to the setting suggested by the reviewer**
>
> We appreciate the reviewer for raising this interesting scenario. When the query is not close to any prompt token, the model can no longer rely on a dominant cluster. Instead, its prediction must be expressed through a combination of multiple feature clusters. This requires extending our analysis in three directions: (i) augmenting the loss decomposition to handle a mixture of several contributing clusters, (ii) generalizing Lemma 2 to track mixture-weighted gradient magnitudes, and (iii) treating training as a sequence of “cluster epochs” in which the dominant set of contributing clusters may change over time. We expect the analysis to be more complex but still manageable with careful arguments and bounding techniques. At the same time, we anticipate that the core mechanisms captured in our work will remain the same. Namely, the curvature $L$ continues to determine the order of attention gradients and therefore the convergence rate. Indeed, **our experiments in Appendix B (e.g., Figures 4-6),** where features are drawn from a continuous Gaussian distribution, empirically confirm that the same curvature-driven behavior persists even when no prompt token is near the query.

---

> ### Author Response · Authors · 2025-11-26
> **Response to Reviewer b5AQ (2/3)**
>
> **Q2**. The experiment results seem not aligned with the theory. The bound of $T_s^\ast$  in Theorem 2 shall be larger than T_f^* in Theorem 1. The authors also claim that "flat regime may enjoy faster convergence" in lines 305-306. However, the convergence of sharp regime in Figure 1(b) is faster than that of flat regime in Figure 1(a).
>
> **A2**: Thank you for pointing this out, and we apologize for the confusion. We clarify that a large Lipschitz constant $L$ can yield faster convergence, with the specific comparison between the flat and sharp regimes depending also on the values of $\Delta$ and $\epsilon$, as explained in Lines 316-320 in our revised manuscript.
>
> To justify the comparison of the two theoretical convergence upper bounds under our experimental setting, in Fig. 1, we used $L=0.1$ for the flat regime and $L=1$ for the sharp regime, with $\Delta=3$ and $\epsilon=0.05$. Plugging these values into Theorems 1-2 yields
>
> $$T_f^\ast=\Theta(Klog(K\epsilon^{-1})/\eta L^2\delta^2\Delta^2)=\Theta(100Klog(0.05K)/9\eta\delta^2)$$
>
> and
>
> $$T_s^\ast=\Theta(Klog(K\epsilon^{-1}\Delta)/\eta\epsilon L^2\Delta^2)=\Theta(20Klog(0.15K)/9\eta\epsilon\delta^2).$$
>
> Thus, the upper bound $T_f^\ast$ under the flat regime is larger than $T_s^\ast$, implying that the sharp regime can converge faster in this configuration, which is consistent with the observation in Fig. 1.
>
> **Q3**. I am not sure how the results are related with the existing linear case analysis. Can the results be reduced to the linear case? Here is a key question, i.e., for the linear case,  could be any arbitrary value <\infinity. However, it seems that previous works do not divide the discussion into two cases like yours. Is it because your analysis is more fine-grained than theirs or anything else? I believe there should be some discussions here.
>
> **A3**: We thank the reviewer for the thoughtful question. Yes, our results naturally reduce to the linear function $f=w^Tx$, where the Lipschitz constant is $L=||w||_2$. Our theory captures how $||w||_2$ governs the convergence behavior depending on whether it lies in a flat or sharp regime. However, existing work on linear ICL (e.g., Huang et al. (2024)) assumes a normalized weight vector $||w||_2=1$, which (together with their feature assumptions) places their setting entirely in our flat regime and thus reflects only that convergence behavior. In contrast, instantiating our framework in the linear setting extends prior results to general $||w||_2$​, further including cases in the sharp regime with different convergence behavior that previous work does not capture.
>
> **Q4**. In Eqn. 4, there is no P in the right-hand side. Why not delete P?
>
> **A4**: Thank you for your observation. We kindly clarify that the P matrix does appear in the right hand of Eq. (4) in terms of its entries of y and X (see definition of P in Eq. (1)). We have clarified this point in our revised manuscript.
>
> **Q5**. Can you theoretically study the case where (1) L is much larger than the case studied in Theorem 2, and/or (2) the noise is not o(1)  in the feature, and/or (3) p_k is not uniform?
>
> **A5**: (1) Theorem 2 already covers the sharp-curvature regime where $L=\Omega(1/(\Delta \delta))$. If $L$ is taken to be even larger, the gradients in Lemma 2 would grow further in magnitude, requiring correspondingly smaller learning rates $\eta$ to avoid instability.
>
> (2) Allowing noise of order $1$ (or larger) would significantly weaken the cluster structure: the query and prompt tokens would no longer concentrate tightly around the underlying feature vectors ${v_k}$, and the query cannot be well-approximated by a single cluster. In this case, attention must aggregate over a much wider set of tokens, making the dynamics substantially more complex. Nevertheless, the core curvature-dependent mechanism we identify, i.e., the Lipschitz constant governs gradient magnitudes and convergence speed, still applies. Please refer to point (2) in our response to Q1 for additional details.

---

> ### Author Response · Authors · 2025-11-26
> **Response to Reviewer b5AQ (3/3)**
>
> (3) Our current theoretical analysis assumes approximately uniform $p_k$ mainly to simplify notation and to highlight curvature-driven effects without introducing additional imbalance terms, but the analysis can be generalized. When the feature frequencies $p_k$ are imbalanced, the attention dynamics can exhibit additional phases that do not appear in the uniform case. If some feature clusters are significantly under-represented, their initial aggregated attention scores $Attn_k$ may be much smaller than those of dominant clusters. This imbalance inflates certain gradient components, particularly those involving interactions between rare and frequent features. Meanwhile, it causes the model to spend an initial phase reducing the influence of over-represented clusters before it can begin to grow the attention weight on the under-sampled feature. As training progresses, the dynamics typically include subsequent phases where the target feature’s self-alignment increases and other off-diagonal interactions are gradually suppressed. These additional stages arise purely from sampling imbalance and primarily affect constants and sample-complexity requirements (e.g., the number of tokens needed to reliably represent rare features). Importantly, the core mechanism highlighted in our analysis remains unchanged: the curvature of the target function continues to dominate the magnitude of attention gradients and therefore continues to govern the convergence behavior.
>
>
> **Q6**. Are there any practical insights from the paper?
>
> **A6**: Although our focus is theoretical, the results offer practical insights for understanding and designing transformer-based in-context learners.
>
> Our analysis in Lemma 2 and Theorems 1-2 shows that the curvature (Lipschitz constant $L$) of the target function directly governs convergence speed: smoother tasks (small $L$) require more iterations, whereas tasks with sharper curvature (large $L$) can be learned rapidly due to larger attention gradients. This provides a principled explanation for why transformers adapt differently to tasks with varying degrees of functional complexity. **This point has been verified in Figure 2 of our experiments.**
> Meanwhile, the identification of two curvature-dependent training regimes suggests that attention behaves differently on tasks with low versus high curvature (in Propositions 2-4). This suggests that task properties, rather than model architecture alone, can dictate the trajectory and speed of in-context learning. **We have verified this insight in deeper transformers in our additional experiments in Appendix B.** Specifically, in Figures 5 and 6, we replace the single-layer model with more realistic two-layer and four-layer Transformers that contain multiple stacked self-attention layers followed by corresponding FFN blocks, while also relaxing the finite feature set assumption by sampling input features from a continuous Gaussian distribution. The results suggest that the core mechanisms identified in our theoretical analysis, which were derived for a single-layer Transformer for analytical tractability, naturally extend to more practical architectures.
>
> **We hope our responses have addressed your questions. If so, we would be sincerely grateful if you would consider raising your rating to reflect this. Of course, we are happy to answer any further questions you may have. Thank you again for the time and thoughtful attention you have given to our work.**

---

### Official Review · Reviewer_DMZQ · 2025-10-28

**Soundness:** 3
**Presentation:** 3
**Contribution:** 3
**Rating:** 6
**Confidence:** 4

**Summary:**

This paper studies the optimization dynamics of Transformers in the setting of in-context learning. Unlike previous work focusing on linear mappings, the authors consider tasks given by $L$-Lipschitz functions, where each input vector in the context is a non-degenerate feature vector perturbed by noise. They show that by optimizing a simplified Transformer’s population risk via gradient descent, the loss can be reduced close to zero. Furthermore, they prove that the convergence rate is governed by the Lipschitz constant $L$, and that two distinct convergence regimes arise depending on the magnitude of $L$.

**Strengths:**

- The paper removes strong assumptions made in prior work (such as linearity or orthogonal feature bases) and establishes theory for a broader class of problems.
- The comparison with previous studies is clear. Existing results on optimization in in-context learning are well summarized, and the paper clearly explains how its contributions differ from them.

**Weaknesses:**

1. The experiments in Section 6 are conducted on a simple case. It would be desirable to evaluate the theory in more practical settings to confirm whether the assumptions and results hold for real-world problems.
2. In addition to data realism, empirical validation on more realistic architectures (e.g., deeper Transformers) would also strengthen the paper.
3. The paper focuses on optimizing population risk, without discussing how the finite-sample training loss landscape might differ or how generalization behaves with respect to sample size.

**Questions:**

1. How would the results change if we considered training the empirical (finite-sample) loss instead of the population risk?
2. The paper’s analysis provides upper bounds parameterized by $L$. While it claims that differences in $L$ lead to distinct convergence behaviors, the tightness of these bounds remains unclear. Can the authors comment on how tight these bounds are?
3. Do the results recover known behaviors in the linear case? Since a linear function $x \mapsto w^\top x$ is also $||w||_2$-Lipschitz continuous, does the Lipschitz constant $||w||_2$- similarly govern convergence?
4. In Figures 1 and 4, the authors claim that different ranges of $L$ lead to differences in convergence speed. While it is indeed observable that larger $L$ tends to result in faster convergence, there does not appear to be a clear separation between the two regimes.
Could the authors provide a more detailed explanation of this point?

---

> ### Author Response · Authors · 2025-11-26
> **Response to Reviewer DMZQ (1/2)**
>
> Thank you for your thorough reviews and constructive comments. We provide our responses to your comments below and have made major revisions in our revised manuscript. To enhance clarity, we have highlighted the revised text in blue for easy identification.
>
> **Q1**. The experiments in Section 6 are conducted on a simple case. It would be desirable to evaluate the theory in more practical settings to confirm whether the assumptions and results hold for real-world problems.
>
> **A1**: We appreciate the reviewer’s suggestion. In Appendix B, we have provided additional experiments for a more practical setting with a continuous feature distribution under non-uniform sampling and larger variability (e.g., Figures 5 and 6). These experiments demonstrate that the core mechanism predicted by our theory, i.e., curvature-dependent gradient scaling and convergence behavior, remains robust even when the data deviate from the idealized model.
>
> **Q2**. In addition to data realism, empirical validation on more realistic architectures (e.g., deeper Transformers) would also strengthen the paper.
>
> **A2**: We thank the reviewer for the constructive suggestion. To address this point, we have added new experiments in Appendix B that evaluate whether our curvature-dependent convergence behavior persists in deeper Transformer architectures. In particular, we replace the single-layer model with two-layer and four-layer Transformers that stack multiple self-attention and FFN blocks, while keeping the data and task setup identical to Figure 4.
>
> As shown in Figures 5 and 6 of Appendix B, these deeper models exhibit the same qualitative dynamics predicted by our theory. We observe the same phase-wise convergence behavior and a clear transition between the flat and sharp L-regimes across all depths. This indicates that the core mechanisms identified in our analysis, although derived for a single-layer model for analytical tractability, extend naturally to more realistic multi-layer architectures.
>
> These additional results strengthen the empirical support for our theoretical findings, and we appreciate the reviewer’s helpful feedback.
>
> **Q3**. How would the results change if we considered training the empirical (finite-sample) loss instead of the population risk?
>
> **A3**: If we replace the population risk by the empirical (finite-sample) loss, the main effect is the introduction of an additional generalization error term on top of the optimization dynamics we analyze. In this setting, we expect the deviation between the empirical and population losses to be bounded by the richness of the prediction function class induced by the attention weight matrix Q, as well as on both the number of sampled tasks (i.e., the number of regression functions) and the number of samples per task. As these quantities (the numbers of tasks and samples) grow, the empirical loss can be shown to uniformly concentrate around the population loss for all Q in the bounded parameter region, and the optimization dynamics become increasingly well-approximated by those derived in our analysis. Once the empirical loss is close to the population loss, we expect that our phase-wise gradient-dynamics results will apply: the curvature-dependent behavior identified in Theorems 1–2 remains unchanged, and the flat/sharp regime separation continues to hold.
>
> **Q4**. The paper’s analysis provides upper bounds parameterized by $L$. While it claims that differences in L lead to distinct convergence behaviors, the tightness of these bounds remains unclear. Can the authors comment on how tight these bounds are?
>
> **A4**: The L-dependence in our bounds is tight up to constant factors. This follows directly from Eq. (2), which imposes the non-degenerate Lipschitz condition $|f(v_k)-f(v_{k'})|=\Theta(L)\cdot \|v_k-v_{k'}\|$. Under this condition, the loss decomposition in Lemma 1 and the gradient magnitudes in Lemma 2 necessarily scale linearly (or quadratically) with L, meaning that the curvature term cannot be improved without strengthening or modifying the problem assumptions.

---

> ### Author Response · Authors · 2025-11-26
> **Response to Reviewer DMZQ (2/2)**
>
> **Q5**. Do the results recover known behaviors in the linear case? Since a linear function x->w^Tx  is also ||w||_2-Lipschitz continuous, does the Lipschitz constant ||w||_2- similarly govern convergence?
>
> **A5**: We thank the reviewer for the thoughtful question. Yes, any linear function $f=w^Tx$ falls into our framework, and our theory captures how $||w||_2$ governs the convergence behavior depending on whether it lies in a flat or sharp regime. However, existing work on linear ICL (e.g., Huang et al. (2024)) assumes a normalized weight vector $||w||_2=1$, which (together with their feature assumptions) places their setting entirely in our flat regime and thus reflects only that convergence behavior. In contrast, instantiating our framework in the linear setting extends prior results to general $||w||_2$​, further including cases in the sharp regime with different convergence behavior that previous work does not capture.
>
> **Q6**. In Figures 1 and 4, the authors claim that different ranges of L lead to differences in convergence speed. While it is indeed observable that larger L tends to result in faster convergence, there does not appear to be a clear separation between the two regimes. Could the authors provide a more detailed explanation of this point?
>
> **A6**: We appreciate the reviewer’s observation. The transition between regimes arises from a smooth change in gradient magnitude rather than a discrete structural shift in the dynamics, and we do not expect a visually sharp separation between them. Instead, the difference appears as a horizontal shift in the curves: larger $L$ leads to fewer iterations to reach a target error. This is because, in both flat and sharp regimes, the attention gradients scale as $\Theta(L^2\Delta^2)$, as shown in Lemma 2. In the sharp regime, we introduce an additional training phase because the gradient magnitude is initially large (due to $L=\Omega(1/(\Delta\delta))$), making the slope decay more apparent and allowing the two stages to be separated experimentally. This behavior is consistent with our theoretical phase-wise characterization. We have clarified in the revised manuscript that the regime change influences iteration complexity rather than producing dramatically different curve shapes, and that the empirical observations align with this interpretation.
>
> **We hope our responses have addressed your questions, especially the additional experiments that illustrate our insights held in more practical experiments and multi-layer transformers. If so, we would be sincerely grateful if you would consider raising your rating to reflect this. Of course, we are happy to answer any further questions you may have. Thank you again for the time and thoughtful attention you have given to our work.**

---

### Official Review · Reviewer_EakK · 2025-10-31

**Soundness:** 3
**Presentation:** 3
**Contribution:** 2
**Rating:** 4
**Confidence:** 4

**Summary:**

This paper studies the training dynamics of attention weights during in-context learning training. It is shown that depending on the Lipschitz constant of the target function class, convergence is either exponential or polynomial. This is confirmed with some empirical evidence.

**Strengths:**

This is a training dynamics analysis, as opposed to characterization of global OPT. The paper is clear to read.

**Weaknesses:**

Features come from a discrete set? The optimal attention pattern is to attend to only exactly identical vectors.

**Questions:**

Take some u such that half of the v_k, those in a set S, satisfy v_k^\top u < 0 (while the other half satisfy v_k^\top u > 0). Consider only functions satisfying f(v_k) = -L for k\in S and f(v_k) = L for k \not\in S. It seems like in this case, there is no reason to attend to only the same feature, and I think you can get this to satisfy Assumption (2). How is his consistent with Attn_k ~ 1?

Why is Attn_k not just |V_k|attn_k?

Should most of these theorems have a “for all k” at the end?

Theorem 1,2 shouldnt there be an upper bound on eta? Usually inverse with smoothness or something.

Isnt Attn_k between 0, 1? So arent both (10) and (11) always positive?

How would this change if you looked at gradient descent on the actual WK, WQ matrices (not Q)?

In the experiments, I think it would be better to have log-scale for the y-axis. This might highlight difference in convergence better (since the difference is something like logarithmic vs polynomial in 1/eps).

---

> ### Author Response · Authors · 2025-11-26
> **Response to Reviewer EakK (1/3)**
>
> Thank you for your thorough reviews and constructive comments. We provide our responses to your comments below and have made major revisions in our revised manuscript. To enhance clarity, we have highlighted the revised text in blue for easy identification.
>
> **Q1**. Features come from a discrete set? The optimal attention pattern is to attend to only exactly identical vectors.
>
> **A1**: We thank the reviewer for raising this point. Our feature model does not assume that input tokens are restricted to a discrete set. As described in Lines 183 and 189 of the original submission (now Lines 187 and 193 in the revised manuscript), each token x is modeled to be a noisy perturbation of an underlying feature vector $v_k$, satisfying $\|x-v_k\|=O(\epsilon_x)$ with $\epsilon_x=o(1)$. In this case, the correct prediction requires non-trivial interpolation across nearby but non-identical features. Our analysis shows that the attention mechanism learns to compensate for these perturbations through curvature-sensitive dynamics (Propositions 2–4), which would not appear in a purely discrete, precisely matching scenario.
>
> **Experimental demonstration:** In Appendix B (Figure 4-6), we further verify experimentally that the predicted dynamics remain valid when tokens are drawn from continuous Gaussian distributions.
>
> Furthermore, even if one interprets the underlying feature set ${v_k}$ as discrete, the resulting learning problem is far from trivial. Unlike exact matching, the model must learn **how nonlinear Lipschitz functions vary across neighboring feature clusters**, which requires attention to capture curvature-dependent differences between outputs $f(v_k)$ and $f(v_{k'})$. These differences scale as $\Theta(L\|v_k-v_{k'}\|)$, creating **feature-dependent prediction gaps** that drive the gradient dynamics (Lemmas 1–2). Moreover, our analysis shows that these gradients undergo a **nonlinear phase transition**: when the Lipschitz constant crosses a threshold, the attention updates switch from a fast-growth regime to a curvature-stabilized slow-growth regime (Propositions 2–4). Establishing convergence under these two regimes requires tight control of (i) how softmax attention amplifies or suppresses feature gaps, (ii) how perturbations propagate into prediction error, and (iii) how curvature contributes multiplicatively to gradient magnitudes. None of these effects appear in prior linear or orthogonal-feature settings, where gradients are curvature-free and attention alignment reduces to simple feature matching.
>
> **Q2**. Take some $u$ such that half of the $v_k$, those in a set $S$, satisfy $v_k^\top u < 0$ (while the other half satisfy $v_k^\top u > 0$). Consider only functions satisfying $f(v_k) = -L$ for $k\in S$ and $f(v_k) = L$ for $k \not\in S$. It seems like in this case, there is no reason to attend to only the same feature, and I think you can get this to satisfy Assumption (2). How is this consistent with $Attn_k ~ 1$?
>
> **A2**: We appreciate the reviewer’s attempt to construct a counterexample. However, the proposed function, where $f(v_k)=-L$ for $k\in S$ and $f(v_k)=L$ for $k\notin S$, does not satisfy our non-degenerate Lipschitz condition in Eq. (2). In Eq. (2), we require that for every feature vector $v_k$, there exists some $v_k’$ such that $|f(v_k)-f(v_{k'})|=\Theta(L) \|v_k-v_{k'}\|$. This condition prevents segment-constant or piecewise-constant functions, because such functions would yield $|f(v_k)-f(v_{k'})|\in{0,2L}$, regardless of the geometric distance $\|v_k-v_{k’}\|$. Such functions therefore do not satisfy our non-degeneracy assumption. The non-degenerate L-Lipschitz condition in Eq. (2) prevents functions that are inherently difficult for gradient-based method to regress, while remaining mild enough to cover as broad a class of target functions as possible.
>
> Under Eq. (2), neighboring features yield different outputs of order $\Theta(L)\cdot \|v_k-v_{k'}\|$, which induces the curvature-dependent gradient terms in Lemma 2. These gradients ensure that the query token increasingly favors tokens sharing the same underlying feature. This yields the regime-dependent dynamics described in Propositions 2–4, where the attention score $Attn_k$ necessarily converges to 1 for the correct feature class. Additionally, the explicit gradient expressions in Eq. (10)-(11) also imply that convergence occurs when $Attn_k^{(t)}=1$. Please see our response to Q6 for more details.
>
> **Q3**. Why is Attn_k not just |V_k|attn_k?
>
> **A3**: We define $Attn_k=\sum_{i:x_i=v_k} attn_k$ in Line 274, which indeed equals $|V_k|attn_k$ when all tokens in $V_k$ share the same feature vector v_k. Our notation simply makes this aggregation explicit and avoids repeatedly carrying both |V_k| and attn_k through the derivations. This aggregated attention score is the quantity that directly determines the prediction output (Eq. (8)) and is thus the natural object to analyze in our theoretical framework.

---

> ### Author Response · Authors · 2025-11-26
> **Response to Reviewer EakK (2/3)**
>
> **Q4**. Should most of these theorems have a “for all k” at the end?
>
> **A4**: Thank you for pointing this out. In our updated version, we have added “for any $k\in[K]$” to Propositions 2-4.
>
> **Q5**. Theorem 1,2 shouldn't there be an upper bound on eta? Usually inverse with smoothness or something.
>
> **A5**: We thank the reviewer for raising this point. In our analysis, we indeed require the learning rate $\eta$ to be smaller than a universal constant (e.g., $\eta<1$) to ensure stability of the gradient updates. This upper bound guarantees that the step size does not distort the order of the curvature-dependent gradients that drive the dynamics. Under this condition, the updates preserve the relative magnitudes established in our smoothness and gradient-growth arguments (see Lemma 8 in Appendix E.2 and Lemma 10 in Appendix F.1). We have made this requirement explicit in Line 235 of the revised manuscript.
>
>
> **Q6**. Isnt Attn_k between 0, 1? So arent both (10) and (11) always positive?
>
> **A6**: Yes, $Attn_k$ lies in [0,1], and therefore both expressions in Eq. (10) and Eq. (11) are non-negative. However, we clarify that Eq. (11) gives the **absolute value** $|g_{k,k’}^{(t)}|$, which does not imply that the underlying gradient $g_{k,k’}^{(t)}$ is always positive. In fact, $g_{k,k’}^{(t)}$ can be either positive or negative depending on the relative values $f(v_k)$ and $f(v_k’)$ under the varying function $f(\cdot)$. This sign fluctuation is precisely what causes $q_{k,k’}^{(t)}$ to oscillate while exhibiting an overall decreasing trend.
>
> By contrast, Eq. (10) shows that $g_k^{(t)}\geq 0$ always holds, ensuring that $q_k^{(t)}$ grow monotonically and drives $Attn_k$ toward $1$ over training. These behaviors are stated explicitly in Propositions 2–3, and the sign and magnitude arguments are detailed in Lemma 4 and Lemma 7 (Appendix E).
>
> Finally, Eq. (10)-(11) also imply that convergence occurs when $Attn_k^{(t)}=1$, in which case both $g_k^{(t)}$ and $g_{k,k’}^{(t)}$ become zero for all $k,k’\in[K]$, marking the fixed point of the dynamics.
>
> **Q7**. How would this change if you looked at gradient descent on the actual WK, WQ matrices (not Q)?
>
> **A7**: Good question! Parameterizing $(W_K, W_Q)$ separately and taking gradient descent over the two matrices lead to more complicated, intertwined dynamics between them, and consequently more complex behavior for the attention scores $Attn_k^{(t)}$ and $Attn_{k,k’}^{(t)}$. One possible way to analyze this added complexity is to derive bounds on the influence of each matrix on the other’s gradient, thereby effectively decoupling their interactions. With such bounds in place, we can characterize their training dynamics in a controlled manner and, in turn, obtain a training dynamics characterization of the attention scores. However, we still expect the **same phase-wise behavior and curvature-dependent convergence patterns** to persist. The gradient expressions in Lemma 2 show that the Lipschitz constant L is one of the dominant terms governing the updates, and this dependence appears regardless of whether we work with the combined matrix $W_{KQ}=W_K^T W_Q$ or treat $W_K$ and $W_Q$ as separate parameters. Thus, the curvature-driven mechanisms identified in our analysis should remain the same under the full parameterization. We also note that in Figures 4-5 of our experiments we use separate $W_K$ and $W_Q$, and the empirical results confirm that the theoretical behavior is preserved.
>
> We also note that this $Q=W_K^TW_Q$ simplification is standard and widely used in theoretical ICL work (e.g., Huang et al., 2024; Yang et al., 2024; Sun et al., 2025), and in many theoretical studies of training dynamics of transformers on other tasks (e.g., [1]-[4] given below).
>
> [1] Zixuan Wang, Eshaan Nichani, Alberto Bietti, Alex Damian, Daniel Hsu, Jason D. Lee, and Denny Wu. "Learning compositional functions with transformers from easy-to-hard data." arXiv preprint arXiv:2505.23683, 2025.
>
> [2] Hongkang Li, Songtao Lu, Pin-Yu Chen, Xiaodong Cui, and Meng Wang. "Training Nonlinear Transformers for Chain-of-Thought Inference: A Theoretical Generalization Analysis." In International Conference on Learning Representations, 2025.
>
> [3] Jianhao Huang, Zixuan Wang, and Jason D. Lee. "Transformers Learn to Implement Multi-step Gradient Descent with Chain of Thought." In International Conference on Learning Representations, 2025.
>
> [4] Yu Huang, Zixin Wen, Aarti Singh, Yuejie Chi, and Yuxin Chen. "Transformers Provably Learn Chain-of-Thought Reasoning with Length Generalization." Advances in Neural Information Processing Systems, 2025.

---

> ### Author Response · Authors · 2025-11-26
> **Response to Reviewer EakK (3/3)**
>
> **Q8**. In the experiments, I think it would be better to have log-scale for the y-axis. This might highlight difference in convergence better (since the difference is something like logarithmic vs polynomial in 1/eps).
>
> **A8**: We thank the reviewer for the suggestion. While a log-scale y-axis is often useful for illustrating exponential decay, in our setting the prediction loss in Fig. 1 does not exhibit a logarithmic relationship with respect to the training epoch $t$. The purpose of these experiments is to verify the dependence of the convergence time on the Lipschitz constant L, namely the $\frac{1}{L^2}$ in the flat regime and $\frac{log(L)}{L^2}$ in the sharp regime (Theorems 1–2). This effect appears primarily as a horizontal shift in the curves rather than a change in vertical decay rate. Using a log-scale on the y-axis would therefore not reveal additional structure and may obscure the comparison across different values of L. We have clarified this point in the revised manuscript.
>
> **We hope our responses have addressed your questions, particularly the clarification of the non-discrete nature of the feature set. If so, we would be sincerely grateful if you would consider raising your rating to reflect this. Of course, we are happy to answer any further questions you may have. Thank you again for the time and thoughtful attention you have given to our work.**

---

### Official Review · Reviewer_FbmK · 2025-11-01

**Soundness:** 3
**Presentation:** 3
**Contribution:** 3
**Rating:** 6
**Confidence:** 3

**Summary:**

The paper investigates the convergence dynamics of a single-layer Transformer trained under the in-context learning (ICL) framework. It establishes convergence guarantees for nonlinear functions in both flat and sharp regimes and reveals a two-phase transition in the evolution of both the loss and the attention scores throughout training.

**Strengths:**

1. The paper studies the convergence of transformers during the pretraining stage, which is an interesting topic.

2. The discovery of the sharp and flat regimes, along with the two-phase transition observed in both the training loss and the attention scores during training, is particularly intriguing.

3. Experimental results show the two-phase transition of the pretraining stage, which coincides with the theory.

**Weaknesses:**

1. The paper focuses only on a single-layer Transformer, and extending the analysis to a multi-layer setting would provide valuable insights.

2. The technical approach appears to overlap with that of [1], which somewhat diminishes the paper’s original contribution.

[1] Huang, Y., Cheng, Y., & Liang, Y. (2023). In-context convergence of transformers. arXiv preprint arXiv:2310.05249.

**Questions:**

1. What are the main technical challenges involved in extending the current framework to a multi-head, multi-layer Transformer architecture?

2. What is the major techinical novelty of extending the linear regression in [1] to the nonlinear regression problem?

[1] Huang, Y., Cheng, Y., & Liang, Y. (2023). In-context convergence of transformers. arXiv preprint arXiv:2310.05249.

---

> ### Author Response · Authors · 2025-11-26
> **Response to Reviewer FbmK (1/2)**
>
> Thank you for your thorough reviews and constructive comments. We provide our responses to your comments below and have made major revisions in our revised manuscript. To enhance clarity, we have highlighted the revised text in blue for easy identification.
>
> **A1**: We thank the reviewer for highlighting this important direction. Extending our analysis to the multi-layer, multi-head setting introduces several substantial technical challenges.
>
> In the **multi-head attention case**, each attention head maintains its own parameter matrix (e.g., $Q^{(h)}$ for head $h$), and attention outputs are aggregated through a shared projection. During training, since each head’s update depends not only on its own attention patterns but also on the evolving outputs of all other heads, the gradients of different heads are tightly coupled. To characterize phase-wise training dynamics as in our paper, one possible approach is to derive bounds on the influence of cross-head weights on each head’s gradients, thereby effectively decoupling interactions among heads. These bounds must be sufficiently tight to capture the true training behavior while still yielding accurate global convergence guarantees. In this way, we obtain a clean decoupling of attention-weight dynamics across heads.
>
> Regarding the **multi-layer architectures**, additional complexity arises from residual connections and intermediate nonlinearities, which cause the query and key embeddings to evolve across layers. As a result, the input to higher layers is no longer the original feature vector but a recursively transformed embedding. This creates a recursive dependency between learned representations and learned attention weights, which is difficult to decouple analytically. Moreover, it requires tracking how attention misalignment and approximation errors propagate through layers, which can accumulate or distort curvature-dependent gradient magnitudes, which is a core aspect of our analysis. Technically, this requires simultaneously controlling (i) the stability of intermediate representations across layers, and (ii) the compounding influence of curvature-dependent gradients, making the analysis far more delicate. To our best knowledge,
>
> **Additional experiments to demonstrate our theoretical insights in multi-layer transformers:** We believe that our current framework lays a useful foundation: the phase-wise dynamics and curvature-sensitive analysis of attention evolution could be extended to multi-layer transformers. To verify this, we have added an additional set of experiments in the Appendix (see Appendix B). In these experiments, we evaluate whether the curvature-dependent convergence behavior persists in deeper Transformer architectures. Specifically, we replace the single-layer model with more realistic two-layer and four-layer Transformers that contain multiple stacked self-attention layers followed by corresponding FFN blocks, while keeping the data and task setup identical to Figure 4. The results show that our key qualitative findings remain consistent: the deeper models exhibit the same phase-wise convergence dynamics, and the flat $L$-regime and sharp $L$-regime phase transition governed by the Lipschitz constant is still clearly observed. This provides empirical evidence that the core mechanisms identified in our theoretical analysis, which are derived for a single-layer model for analytical tractability, extend naturally to deeper, more realistic Transformer architectures.
>
> We view the theoretical extension as a promising direction for future work and have briefly discuss it in the revised manuscript.

---

> ### Author Response · Authors · 2025-11-26
> **Response to Reviewer FbmK (2/2)**
>
> **A2**: We thank the reviewer for raising this concern. While our analysis is inspired by the general training-dynamics framework of Huang et al. (2023), the technical challenges of **nonlinear** regression that our paper addresses do not appear in the **linear** regression setting studied in [1]. Our work, to the best of our knowledge, is the first to show that the Lipschitz constant itself fundamentally governs (i) gradient scales, (ii) phase transitions between training regimes, and (iii) the resulting convergence time. In this sense, our analysis is more fine-grained than prior linear work: it strictly generalizes the linear case while revealing new phenomena that don’t exhibit in linear regression. were previously hidden by normalization.
>
> Our main technical novelty arise precisely from this nonlinear structure:
>
> - **Curvature-dependent loss decomposition (Lemma 1):** We derive a new loss decomposition that explicitly ties the prediction error to attention weights and function curvature via Lipschitz-induced feature gaps. Such curvature-sensitive decomposition does not exist in the linear setting of [1].
>
> - **Curvature-Governed Gradient Dynamics (Lemma 2 and Propositions 2–4):** We further characterize how **function curvature** (quantified by the **Lipschitz constant**) governs the magnitude and evolution of attention gradients (Lemmas 2, 4, and 7 in Appendices E.1 and E.2). We then establish two distinct training regimes, i.e., flat vs. sharp curvature, each with provably different convergence behaviors (Propositions 2, 3, and 4), revealing a phase transition in attention dynamics. This phenomenon is absent in [1], where gradients do not depend on curvature and therefore cannot exhibit regime shifts.
>
> - **Curvature-Sensitive Convergence Guarantees:** Our convergence rates depend explicitly on the function’s Lipschitz constant $L$, revealing how nonlinear curvature accelerates or slows attention concentration. In contrast, [1] provides curvature-free, linear-task results that cannot capture function-dependent dynamics.
>
> In summary, the techniques, lemmas, and convergence arguments we develop are fundamentally new and are very different from those in [1]. Meanwhile, the linear function $f=w^Tx$ studied in [1] falls into our framework as a special case, where $L=||w||_2$ is the Lipschitz constant. When $||w||_2$ lies in the flat regime, the training dynamics exhibit the same two-phase structure, i.e., fast initial growth followed by steady refinement, under the balanced feature set of Eq. (7). This aligns with the qualitative behaviors established in [1]. However, we note that [1] assumes a normalized weight vector (e.g., $||w||_2=1$) precisely to remove the influence of $||w||_2$ from the gradient magnitudes. In contrast, our work is the first to characterize how the Lipschitz constant itself (here $||w||_2$) governs gradient scales, phase transitions, and convergence time. Thus, our framework not only recovers the linear setting but also extends it by revealing how curvature/Lipschitzness fundamentally shapes attention-based learning dynamics.
>
> **We hope our responses have addressed your questions, particularly the new experiments we provided to demonstrate the broader effectiveness of our insights. If so, we would be sincerely grateful if you could consider raising your rating to reflect this. Of course, we are happy to answer any further questions you may have. Thank you again for the time and thoughtful attention you have given to our work.**

---

### Meta-Review · Area_Chair_P5mD · 2026-01-09

**Summary:**

This submission analyzes the in-context learning of a class of nonlinear functions using shallow transformers optimized by population gradient. The reviewers raised the following concerns.

* Reviewers b5AQ, DMZQ, and FbmK complained about the limited theoretical setting (one-layer attention, simple function class, population gradient, etc.). The authors partly addressed this concern by explaining the novelty and technical challenges in handling the nonlinear function class they studied.

* Reviewers b5AQ, DMZQ, and EakK questioned the clarity and scope of the empirical validation. In the revision, the authors included additional experiments on deeper architectures and also probed the robustness of the observed phenomenon.

* Reviewers b5AQ and FbmK asked about overlap with prior results on linear ICL. The authors clarified that prior works do not capture the curvature-dependent convergence behavior.

Overall, the authors' response partly addressed the reviewers' concerns, hence placing the submission on the borderline.

**Reviewer Concerns:**

See above.

**Reviewer Scores:**

Reviewers b5AQ and EakK both gave a score of borderline reject. Their concerns (in particular, the theoretical contribution and novelty) are partly addressed, and the area chair believes that one of them might increase their evaluation to borderline accept. The other two reviewers are likely to maintain their score of borderline accept.

---

### Decision · Program_Chairs · 2026-01-26

Reject